# FRA-1 as a Regulator of EMT and Metastasis in Breast Cancer

**DOI:** 10.3390/ijms24098307

**Published:** 2023-05-05

**Authors:** Laura Casalino, Francesco Talotta, Ilenia Matino, Pasquale Verde

**Affiliations:** Institute of Genetics and Biophysics “A. Buzzati Traverso”, Consiglio Nazionale delle Ricerche (CNR), Via Pietro Castellino, 111, 80131 Naples, Italyilenia.matino@igb.cnr.it (I.M.)

**Keywords:** AP-1 transcription factors, FRA-1, *FOSL1*, TNBC, EMT

## Abstract

Among FOS-related components of the dimeric AP-1 transcription factor, the oncoprotein FRA-1 (encoded by *FOSL1*) is a key regulator of invasion and metastasis. The well-established FRA-1 pro-invasive activity in breast cancer, in which *FOSL1* is overexpressed in the TNBC (Triple Negative Breast Cancer)/basal subtypes, correlates with the FRA-1-dependent transcriptional regulation of EMT (Epithelial-to-Mesenchymal Transition). After summarizing the major findings on FRA-1 in breast cancer invasiveness, we discuss the FRA-1 mechanistic links with EMT and cancer cell stemness, mediated by transcriptional and posttranscriptional interactions between *FOSL1*/FRA-1 and EMT-regulating transcription factors, miRNAs, RNA binding proteins and cytokines, along with other target genes involved in EMT. In addition to the FRA-1/AP-1 effects on the architecture of target promoters, we discuss the diagnostic and prognostic significance of the EMT-related FRA-1 transcriptome, along with therapeutic implications. Finally, we consider several novel perspectives regarding the less explored roles of FRA-1 in the tumor microenvironment and in control of the recently characterized hybrid EMT correlated with cancer cell plasticity, stemness, and metastatic potential. We will also examine the application of emerging technologies, such as single-cell analyses, along with animal models of TNBC and tumor-derived CTCs and PDXs (Circulating Tumor Cells and Patient-Derived Xenografts) for studying the FRA-1-mediated mechanisms in in vivo systems of EMT and metastasis.

## 1. Introduction

Breast cancer represents the second most common cancer and the most frequent cancer among women worldwide. During recent years, the histopathological classification of the clinically heterogeneous mammary gland adenocarcinomas has been complemented by transcriptional profiling of tumor specimens. The association of specific expression arrays with distinct subtypes defines signatures that classify breast cancer subtypes (Figure 1) and provides novel diagnostic and prognostic biomarkers. Based on the expression profile of 50 selected genes, the widely used PAM50 classifier allows to recognize five intrinsic subtypes of breast cancer: luminal A; luminal B; HER2-enriched; normal-like; and basal-like, in increasing order of grading, proliferation, and expression of basal-like genes [1,2]. According to a different classification (surrogate intrinsic subtypes) mainly based on histology and IHC detection of key biomarkers, the TNBC (Triple-Negative-Breast-Cancer) subtype is characterized by the absence of ER and PR expression along with the lack of HER-2-amplification. The basal-like subtype is commonly referred to as TNBC, although not all basal-like are triple-negative, and vice versa. In contrast with the antiestrogen-mediated treatment of ER-positive adenocarcinomas and the trastuzumab-mediated treatment of HER-2-amplified cancers, no targeted therapy is available for TNBC, which is responsible for most breast cancer-associated deaths [3]. The 5-year survival rate is less than 30%, despite the adjuvant chemotherapy that usually follows surgical excision. The clinical course, generally associated with lymph node involvement at the time of diagnosis, reflects the aggressive nature of the TNBC [4,5].

TNBC clinical aggressiveness and chemoresistance correlate with the activation of pro-invasive programs implicated in metastatic dissemination and gain of stem-like features, which fuel the compartment of therapeutically resistant CSCs (Cancer Stem Cells). Gene expression profiling shows that TNBCs can be further distinguished into six different subtypes: BL1 (Basal-Like 1); BL2 (Basal-Like 2); ML (Mesenchymal-Like); MSL (Mesenchymal/Stem-Like); IM (Immuno-Modulatory); and LAR (Luminal Androgen Receptor). The highly dedifferentiated claudin-low subtype includes the ML and MSL subtypes [3,5,10,11,12].

In addition to the low expression of claudins, contributing to the loss of epithelial tight junctions, the claudin-low subtype exhibits several hallmarks of EMT (Epithelial-to-Mesenchymal Transition), associated with the gain of stemness along with high levels of MAPK pathway activation and stromal infiltration [13].

Noteworthy, most in vitro analyses on invasive breast cancer, including most studies summarized in the present review, rely on a group of TNBC-derived cell lines (MDA-MB-435, MDA-MB-436, Hs578T, BT549, MDA-MB-231, MDA-MB-157, SUM1315MO2, SUM159PT, and HBL100), which exhibit gene expression patterns similar to the claudin-low subtype [14].

The claudin-low phenotype, which is associated with the increased fraction of chemoresistant CSCs, can be induced in morphologically epithelial MMC (Mouse Mammary Carcinoma) cells and human preneoplastic cells (MCF10A) by treatment with the well-characterized EMT inducers, such as TGF-beta and TNF-alpha [15]. Both cytokines play key roles in the crosstalk between cancer cells and multiple cell types recruited to the tumor microenvironment (fibroblasts, macrophages, immune cells, mesenchymal stem cells, etc.).

In response to paracrine cues, the EMT transcriptional programs are controlled by six “core EMT-TFs” (SNAI1/SNAIL, SNAI2/SLUG, TWIST1, TWIST2, ZEB1, ZEB2), which act in a partially redundant tumor-specific manner to inhibit the expression of epithelial genes and induce the pro-invasive mesenchymal transcriptional programs (for a recent review, see [10]). Along with the core EMT-TFs, an increasing number of transcription factors (e.g., PRRX1, FOXC2, SIX1, YAP/TAZ) are involved in the control of EMT and metastasis in breast cancer cells (BCCs). The EMT-TFs are assisted by a variety of epigenetic modifiers (e.g., PRC1/2, NuRD, LSD1, SUV93H1/2) that play a major role in the plasticity of EMT. Multiple epigenetic states are associated with intermediate phenotypes between the stably epithelial and stably mesenchymal cell states (e.g., the metastable quasi-epithelial vs. quasi-mesenchymal phenotypes) [16]. In addition, the epigenetic reader of Lys-acetylation BRD4 is critically implicated in the positive regulatory function of EMT-TFs, such as TWIST, in TNBC cells [17,18].

The epigenetic mechanisms of EMT also involve the miRNA-mediated posttranscriptional regulation of EMT-TFs and other key EMT regulators (SIRT1, SUZ12, HMGA1/2), which, in turn, repress the transcription of the cognate miRNAs. The resulting network of interconnected double-negative feedback loops is critically implicated in the maintenance of the epithelial/mesenchymal phenotypes of the BCCs [19].

The posttranscriptional control of EMT in breast cancer depends on multiple factors, including a wide range of miRNAs, RNA-binding proteins (RBPs), and RNA modifications, which affect the key steps of RNA processing, including editing, splicing, transport, intracellular localization, and translation regulation (reviewed in [20,21]). RBPs differently affect the expression and function of tumor-related transcripts and can act as tumor suppressors or promoters. For instance, the TGF-beta-induced EMT in malignant BCCs requires the downregulation of RBM38 (RNA Binding Motif Protein 38) mediated by Snail and c-Myc through their direct binding to an E-box element in the promoter region [22]. Similarly, QKI (Quaking) is downregulated in different breast tumor subtypes (RBM38 ER, PR, and HER2 positive, non-basal-like, and non-TNBC) while PCBP2 (Poly (C) binding protein 2), HuR (Hu-antigen R), LIN28, SAM68, and MSI-1 and -2 (Musashi-1 and -2) are upregulated [20]. The regulatory targets of RBPs include several key oncosuppressors (ZO-1 -Zonula occludens-1- and PTEN), oncogenes (p53 and c-Myc), and EMT-TFs (SNAIL and CBP/beta-catenin). RBPs also modulate breast cancer invasion and metastasis by affecting extracellular proteases (uPAR, MMP-9), angiogenic factors (VEGFA), oncogenic signaling pathways (PI3K/Akt/Gsk-3β, RASA, and MAPK, *Hippo* pathway) or promoting the competing endogenous RNA (ceRNA) network crosstalk among STARD13, CDH5, HOXD10, and HOXD1 (STARD13-correlated ceRNA network) [20].

Here we will summarize the regulatory interactions and roles played by the AP-1 component FRA-1 in the control of EMT in invasive breast cancer.

## 2. The AP-1 Family of Transcription Factors

The AP-1 (Activator Protein-1) complex is assembled as a dimer between members of the bZIP (basic-leucine zipper) subfamilies JUN and FOS, along with MAF (Musculoaponeurotic Fibrosarcoma) and ATF (Activating Transcription Factors). The JUN-family members (c-JUN, JUNB, and JUND) can form JUN homodimers and stable JUN/FOS heterodimers with the FOS proteins (c-FS, FOSB, FRA-1, and FRA-2), which can only heterodimerize with JUN proteins. The dimerization depends on the coiled-coil association of two α-helices, which is mediated by hydrophobic contacts between the heptad repeats of leucine residues within the bZIP region.

The AP-1 components are transcriptionally and posttranscriptionally controlled by a large variety of extracellular stimuli, including growth factors, cytokines, stress, infection, and oncogenic pathways. AP-1 dimers bind to the TPA-response element (TGAG/CTCA) and similar motifs, predominantly localized in distal enhancers rather than proximal promoters [23]. The AP-1 complex controls a wide range of target genes implicated in many, if not all, cellular functions, including proliferation, apoptosis, autophagy, differentiation, migration, and invasion [24,25].

The *JUN* and *FOS* proto-oncogenes were discovered as cellular homologs of retroviral oncogenes *v-Jun* and *v-Fos*. In concert with (or antagonistically to) c-JUN and c-FOS, distinct JUN and FOS family members differentially affect cell cycle progression and apoptosis [26], and several of them, such as *JUNB*, *JUND*, *FOS*, and *FOSB*, exert context-dependent oncogenic or oncosuppressor roles [27]. Accordingly, in a seminal review article [25], the AP-1 complex has been defined as a “double-edged sword” in tumorigenesis.

AP-1 is a highly dynamic regulator of gene expression whose outcomes are shaped by the nature and duration of multiple signaling pathways. The cell context-dependent positive or negative regulation of specific target genes results from multiple regulatory layers, including the relative abundance and posttranslational modifications of individual components, dimeric composition, and interactions with other nuclear proteins, along with various factors affecting the AP-1 binding and transactivation mechanisms. As recently summarized [23], the integration between various “omics” technologies has begun to shed light on the complex networks of AP-1 interactions with transcriptional enhancers, chromatin domains, and DNA-binding partners.

### 2.1. FRA-1 Structure and Regulation

The FOS-family protein FRA-1 (Fos-Related-Antigen 1), named because of immunological cross-reactivity with c-FOS, is encoded by the *FOSL1* (*FOS*-Like-1) gene on chr11q13. The 271-aa-long FRA-1 oncoprotein is one of the most frequently overexpressed FOS-family members in a large variety of solid tumors. The oncogenic roles and regulation of *FOSL1*/FRA-1 in tumorigenesis have been described in several recent exhaustive reviews [28,29,30,31].

Various transcriptional, posttranscriptional, and posttranslational mechanisms are implicated in FRA-1 accumulation in response to multiple oncogenic lesions. The RAF-MEK-ERK, IL6-Stat3, and Wnt-beta-catenin signaling pathways induce the *FOSL1* transcription through multiple regulatory sites localized in both 5′ flanking and intronic enhancer elements. The mechanisms of c-MYC binding to the *FOSL1* enhancer, triggering the chromatin changes associated with the sequential recruitment of multiple histone modifiers and readers [32,33], have been recently reviewed with other aspects of *FOSL1* transcriptional and posttranscriptional regulation [28,29].

Recently, the c-MYC-dependent control of *FOSL1* in breast cancer has been further characterized by studying the NRG-dependent control of c-MYC stability. The NRG-ERK1/2-FBXW7-c-MYC pathway is responsible for the ERK-induced recruitment of c-MYC to the *FOSL1* promoter in TNBC. Being overexpressed in about 30% of HER2-negative breast cancers, the EGF family member Neuregulin/NRG1 is pathogenetically relevant in TNBC. NRG1 activates the MAPK pathway, and the ERK1/2-mediated phosphorylation of the ubiquitin ligase FBXW7 results in decreased polyubiquitylation and increased nuclear import of c-MYC, which induces FRA-1 accumulation and lung metastasis in vivo [34].

In addition to transcriptional induction, the cancer-associated FRA-1 overexpression depends on the downregulation of the oncosuppressor miRNAs, which target the *FOSL1* transcript [28,29]. Remarkably, in breast cancer progression, the downregulation of miRNAs contributes to FRA-1 accumulation not only in neoplastic cells but also in tumor-associated cell types, such as TAMs (Tumor-Associated Macrophages). Relevant examples include miR-34 and miR-130a, which suppress breast cancer invasion and metastasis by targeting FRA-1 in cancer cells [35,36], while miR-19a-3p and miR-4516 affect cancer cell invasiveness by targeting FRA-1 in TAMs [37] and CAFs (Cancer-Associated Fibroblasts) [38].

Along with oncomiRs, the posttranscriptional control of *FOSL1* involves RNA modifications and RNA-binding proteins. The cancer-associated epitranscriptome provides a new perspective in deciphering tumor progression. Dynamic RNA modifications involve the crosstalk among writers, erasers, and readers [39]. The most prevalent mRNA modification, *N*^6^-methyladenosine (m^6^A), affects the stability and translation of key oncogenic transcripts. The effect of the m^6^A modification on *FOSL1* mRNA stability is suggested by various findings, including a recent analysis of the m^6^A methylome in response to triptolide in rheumatoid arthritis, in which *FOSL1* mRNA methylation and abundance is strongly affected, along with IGF2BP3, encoding of the insulin-like growth factor 2 mRNA-binding protein 3 [40]. Interestingly, IGF2BP1/2/3 is a major m^6^A reader, mechanistically linking mRNA methylation and stability. These RBPs are critically involved in posttranscriptional control of stemness and neoplastic transformation by targeting thousands of m^6^A-modified mRNA transcripts [41]. A direct role in EMT has been proven for IGF2BP1 and IGF2BP3, implicated in the SNAI2/SLUG-dependent gain of mesenchymal features through the direct association of IGF2BP3 (IMP3) with the *SNAI2/SLUG* mRNA [42] or the LEF1-mediated transactivation of *SNAI2/SLUG* resulting by the IGF2BP1 association with the *LEF1* mRNA [43].

Interestingly, one of the publicly available databases (RIP-chip GeneST from ENCODE, accessible through UCSC Genome Browser) shows that the *FOSL1* mRNA is among the transcripts bound by IGF2BP1 (our unpublished observation).

Other RNA-binding proteins might cooperate with IGF2BP1/2/3 in posttranscriptional control of *FOSL1*. A likely candidate is represented by PTPB1 (RNA Polypyrimidine Tract-Binding Protein 1), recently implicated in the mechanism of FRA-1-mediated *PD-L1* transcription, immune escape, and metastasis in response to the abnormal expression of HOXA11-AS1 lncRNA (long-noncoding RNA) in one subtype (hypopharyngeal carcinoma) of HNSCC (head and neck squamous cell carcinoma). In this system, the *FOSL1* mRNA half-life is positively controlled by the direct interaction with PTPB1, which is enhanced by the HOXA11-AS1 lncRNA [44]. Interestingly, the paralogous factor PTPB3, which prevents the *ZEB1* mRNA degradation by binding to its 3′UTRis, contributes to the EMT posttranscriptional regulation in breast cancer [45].

Multiple posttranslational modifications control the FRA-1 stability, DNA binding, and transactivating activity. The MEK/ERK/Rsk and PKC-theta pathways control the FRA-1 protein stability and DNA binding activity via a cascade of phosphorylation events on serine/threonine residues in response to numerous oncogenes, cytokines, and growth factors. As recently reviewed [29], in non-transformed cells, FRA-1 is an intrinsically unstable, short-lived protein, while in invasive cancer cells, the increased activity of the RAS/RAF/MEK/ERK pathway induces the phosphorylation of FRA-1 residues S252 and S265, which inhibit the C-terminal (DEST) domain implicated in FRA-1 proteasomal degradation [46]. In invasive breast cancer, PKC-theta contributes to both FRA-1 stabilization, via T223 and T230 phosphorylation, and transactivation activity, through the FRA-1 phosphoacceptor residues T217 and T227 [47,48]. Moreover, the RAS-independent Mixed-lineage-kinase 3 contribute to FRA-1/AP-1 accumulation by inducing the activity of both JNK and ERK, responsible for the phosphorylation-mediated stabilization of c-JUN and FRA-1, respectively [49].

### 2.2. FRA-1 in Tumorigenesis

FRA-1 overexpression has been implicated in cancer cell proliferation, survival, migration, invasion, and plasticity in a large variety of neoplastic diseases, including most solid tumors. FRA-1 oncogenic roles have been discussed in a number of relevant reviews, dealing with many aspects of FRA-1 in tumorigenesis and metastasis mechanisms, along with recently proposed strategies of FRA-1 therapeutic targeting [28,29,30,31,50,51,52,53,54,55].

Regarding EMT mechanisms, the FRA-1-mediated control of cell plasticity and cancer-associated embryonic signaling pathways have been addressed in a highly relevant review, dealing with the FRA-1 involvement in EMT regulatory networks [28].

In the present review, we will focus on the functional roles, transcriptional mechanisms, and clinical significance of *FOSL1*/FRA-1 in the control of EMT and metastasis in invasive breast cancer.

### 2.3. FRA-1 Controls Breast Cancer Cell Motility, Invasion, and Proliferation

Seminal evidence on the link between Fra-1 overexpression and the hallmarks of EMT was obtained in a mouse mammary cancer cell system. Comparison between the parental cell line and the metastatic derivative showed the mutually exclusive expression of Fra-1 and E-cadherin, which was supported by the analysis of multiple human adenocarcinoma cell lines. Moreover, FRA-1 ectopic expression could trigger cancer cell motility and invasion, along with the accumulation of various metastasis-associated proteins, such as HMGA1, S100A4, and components of the plasminogen activation system (uPA, uPAR, PAI-1) [56].

Complementarily, the pioneering expression profiling of a panel of (weakly vs. highly invasive) human breast cancer cell lines resulted in a 24-genes signature which highlighted the FRA-1 overexpression along with its tight association with vimentin upregulation in the phenotypically mesenchymal cell lines [57].

Subsequent functional studies were based on FRA-1 ectopic expression in the non-invasive ER-positive MCF7 cells vs. FRA-1 knockdown in the invasive MDA-MB-231 cells. Along with motility and invasion, FRA-1 drove the secretion of metalloproteases and VEGF, and FRA-1 overexpression/downregulation affected breast cancer cell proliferation and cyclin D1 expression [58].

## 3. FRA-1 Is Required for EMT and Gain of Stem-like Features

The above functional studies highlighted the pivotal FRA-1 function in the gain of key mesenchymal properties, such as motility and invasiveness in human tumor cell lines. Independent lines of investigation highlighted the FRA-1 effects on EMT markers (E-cadherin and vimentin) in response to the overexpression of the RAS oncoprotein or oncogenic protein-kinases in immortalized breast epithelial cells. The HRASV12-mediated induction of EMT in MCF10A cells, in which ERK2- but not ERK1- is implicated in the gain of mesenchymal features, required FRA-1 expression. In addition, FRA-1 was necessary for the RAS- and ERK2-induced accumulation of ZEB1, thus pointing to the RAS-ERK2-FRA-1-ZEB1 pathway in the control of EMT [59,60].

In a rat kidney cell system, in which the tyrosine kinase receptor TrkB triggered in vitro motility and invasiveness along with lung metastasis of tumor xenografts, *Fosl1* was identified among the top overexpressed genes. Expression profiling and functional analyses were extended to a prototypical TNBC human cell line (MDA-MB-231), which was compared to a highly metastatic derivative. FRA-1 knockdown fully rescued the E-cadherin, downregulated vimentin expression, and suppressed the lung metastases arising in vivo after tail vein injection [7].

Based on the key findings in the developing mammary gland and breast cancers [61], the relationship between the EMT-associated cancer cell plasticity and the gain of stem-like features has been established in a variety of normal and neoplastic tissues (reviewed in [62]).

FRA-1 was proposed to act as the “gatekeeper” of the mesenchymal transition associated with the gain of stem-like features in the human cell system of NAMECs (naturally arising mesenchymal cells), deriving from human mammary gland epithelial cells (HMLE). In these cells, the mesenchymal transition is associated with changes in AP-1 dimeric composition in response to the interplay between distinct signal transduction pathways. Activation of the EMT program coincides with the transition from the EGF-EGFR-c-FOS pathway in the non-CSCs, to the PDGF-PDGFR-PKC-alpha-FRA-1 pathway in CSCs. Thus, the AP-1 compositional change from JUNB/JUND-c-FOS heterodimers to c-JUN-FRA-1 heterodimers contributes to the EMT triggered by PKC-alpha [19].

In addition to PKC-alpha, FRA-1 expression is regulated by the less characterized PKC-theta isoform. FRA-1 residues are directly or indirectly phosphorylated by PKC-theta, with diverse consequences on FRA-1 activity in TNBC cells. While FRA-1 phosphorylation on S265, T223, and T230 strongly increases the protein stability [47], phosphorylation of T217 and T227 mainly affects the FRA-1 transcriptional activity [48]. Interestingly, the chromatin-associated form of PKC-theta has been implicated in EMT and stemness in BCCs [63] through epigenetic mechanisms at least partially mediated by the phosphorylation and activation of the histone demethylase LSD1 [64]. The selective enrichment and colocalization with beta-catenin and other EMT markers of the PKC-theta-phosphorylated protein (FRA-1-phospho-T217) at the tumor stroma interface suggest that FRA-1 participates in the gain of mesenchymal features and stemness in response to PKC-theta in invasive breast cancer [48].

Although the BRAF-MEK1-ERK2 is the best-studied pathway responsible for FRA-1 accumulation [28,46,59,60], other MAPKKKs have been implicated in the FRA-1-mediated control of invasion and vascular intravasation in TNBC cells. MLK3 (Mixed Lineage Kinase 3) is involved in oncogenesis through multiple mechanisms, mainly involving the JNK-mediated phosphorylation of paxillin and consequent turnover of focal adhesions. MLK3 has been implicated in FRA-1 accumulation through JNK- and ERK-mediated mechanisms affecting the FRA-1 mRNA expression level. Interestingly, in addition to the FRA-1-mediated induction of MMP1, MMP9, and ECM invasion, the same study highlights the role of the MLK3-FRA-1-MMPs axis in the increase in endothelial permeability, which is essential for vascular intravasation by the TNBC cells undergoing the first steps of metastatic dissemination [49].

### 3.1. FRA-1 Is Both a Transcriptional Inducer and Target of Multiple EMT-TFs

Several lines of evidence from multiple cell systems show that FRA-1 participates in the network of regulatory interactions between the major EMT-TFs. The major results, inferred from investigations in various cell systems, are summarized in Figure 2.

First, in a study on the MAPK-dependent control of cell migration in multiple TNBC lines, the EMT-TF SLUG (SNAI2) was essential for in vitro invasion and lung metastasis and expressed in an AP-1-dependent manner, in response to the ERK-Fra-1/c-JUN axis [65].

In the NAMEC cell system, FRA-1 expression is essential for EMT, not only in spontaneously transitioning cell cultures but also in cell lines ectopically expressing the tamoxifen-inducible TWIST and SNAIL derivatives. FRA-1 accumulation results from direct transactivation by both EMT-TFs binding to two *FOSL1* regions around the TSS and within the first intron. In addition, FRA-1 expression is required for both TWIST-induced expression of SNAIL and SNAIL-induced expression of TWIST. Therefore, FRA-1 is both a transcriptional target and a regulator of both TWIST and SNAIL. Importantly, FRA-1 was required for the TWIST- or SNAIL-mediated induction of *ZEB1*, *ZEB2*, and *SLUG* [19].

The mechanistic links between FRA-1 and *ZEB1/2* have been further elucidated in subsequent analyses of the FRA-1 cistromes and transcriptomes in both human and mouse cell systems, in which the FRA-1-regulated transcriptional networks were dissected through ChIP-seq analyses combined with expression profiling of FRA-1 (and c-JUN) target genes [8,9,66].

**Figure 2 ijms-24-08307-f002:**
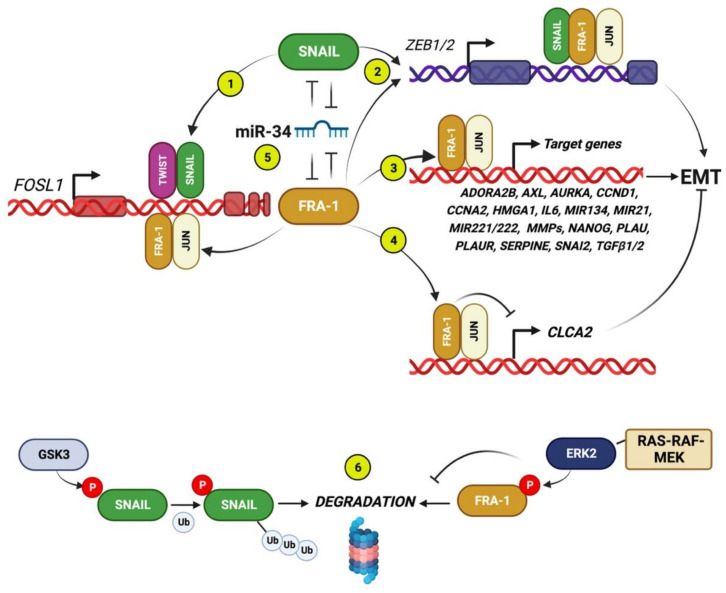
FRA-1 interactions with core EMT-TFs, mRNAs, and EMT-related target genes. Circled numbers indicate the following regulatory interactions. (1) *FOSL1* is transcriptionally induced by SNAIL (and TWIST), binding to the *FOSL1* intronic enhancer [19]. (2) The FRA-1/AP-1 heterodimers cooperate with SNAIL for transactivating ZEB1/2 [19]. (3) In addition to core EMT-TFs, the FRA-1/AP-1 dimers positively control many target genes encoding for multiple components implicated in EMT in breast cancer: cytokines (*IL6*, *TGFB1/2*); GPI-anchored (*PLAUR*); G-protein-coupled (*ADORA2B*) and tyrosine kinase receptors (*AXL*); extracellular proteases (*MMPs*, *PLAU*, *SERPINE1*); mitosis-regulating protein-kinases (*AURKA*); cyclins (*CCND1*, *CCNA2*); non-histone chromatin components (*HMGA1*); and miRNAs [7,8,9,66]. (4) The FRA-1/AP-1-mediated repression of *CLCA2* downregulates the expression of the calcium channel component CLCA2 [8], which also participates in cell-cell junctions, thus representing a negative EMT regulator, as discussed in Section 3.4. (5) The double-negative feedback loops formed by the *FOSL1* and *SNAI1* transcripts, targeted by the same oncosuppressor miRNA family (miR-34), which is a strong EMT inhibitor in BCCs. (6) In addition to the above illustrated transcriptional and posttranscriptional interactions, both FRA-1 and SNAIL are posttranslationally controlled by phosphorylation: while the ERK2-mediated phosphorylation stabilizes FRA-1 in response to the RAS-RAF-MEK pathway, GSK3-beta positively controls SNAIL ubiquitylation and degradation [67].

The first direct evidence of Fra-1 as a transactivator of both *Zeb1* and *Zeb2* promoters was obtained in a mouse mammary epithelial cell (MMLE) system. In the fully polarized non-tumorigenic cell line EpH4, FRA-1 overexpression triggered the gain of mesenchymal features associated with in vivo tumorigenicity and formation of metastasis in orthotopic xenograft models. ChIP analyses and reporter assays showed the Fra-1 functional association with the *Zeb1* first intron and the *Zeb2* promoter region. Accordingly, the *Zeb1* or *Zeb2* knockdown restored the epithelial features in the Fra-1-overexpressing EpH4 cells [66].

According to the key role played by posttranslational control of EMT-TFs accumulation, the RAF-MEK-ERK2-mediated FRA-1 stabilization is represented, along with the GSK3-mediated SNAIL phosphorylation and destabilization [67] (Figure 2).

The mechanisms implicated in the FRA-1/AP-1-mediated control of ZEB2 expression and activity are summarized in Figure 3.

According to the FRA-1 cistrome, investigated in the TNBC cell line BT549, most FRA-1 binding sites are shared with c-JUN, thus indicating a major role played by c-JUN/FRA-1 heterodimers in the FRA-1-dependent transcriptional control [8]. FRA-1 (along with c-JUN) accumulation is required for the induction of EMT in response to TNF-alpha through mechanisms involving the FRA-1-mediated transactivation and chromatin looping of the *ZEB2* promoter region [68].

In addition to *ZEB2* transcriptional induction, resulting in ZEB2 accumulation, the FRA-1/c-JUN dimers also control the ZEB2 activity (Figure 3), inhibiting the GATA-family transcriptional repressor TRPS1 via indirect mechanisms dependent on the FRA-1/AP-1-mediated induction of miR-221/222, as discussed below (Section 3.4).

In addition to the reciprocal transcriptional controls with the core EMT-TFs, FRA-1 participates in the breast cancer-associated EMT by further mechanisms, including several protein–protein interactions between FRA-1 and key TFs on specific subsets of regulatory targets.

The key interaction between FRA-1/AP-1 and YAP/TAZ complexes emerges from various lines of evidence. First, the frequent co-localization of the FRA-1 and c-JUN binding sites with the TEAD binding motif suggested the regulatory interaction between FRA-1/AP-1 and the *Hippo* pathway on target promoters [8]. The functional cooperation between AP-1 and the *Hippo* pathway was further highlighted by a study in which the TEAD coactivator YAP1 was identified among the genes able to rescue the effect of KRAS suppression in KRAS-transformed colorectal cancer cells. In this context, FOS expression was essential for both KRAS and YAP1 effects on cell viability. The c-FOS interaction with YAP1 was required for the expression of the EMT-related genes SLUG (SNAI2) and VIM through direct binding to their respective promoters. However, the relative contribution of FRA-1 was not addressed in this study [70].

In TNBC cells, in-depth analyses of the *Hippo* downstream effectors showed that the YAP/TAZ-driven transcription program is critically involved in the control of cell proliferation. Genome-wide analyses of YAP, TAZ, and TEAD binding site distribution highlighted the co-occupancy with AP-1 on transcriptional enhancers. c-JUN was present on almost 80 percent of the YAP/TAZ/TEAD binding sites, and, in all the analyzed binding sites, FRA-1 was bound along with c-JUN, thus suggesting its role as a major c-JUN heterodimeric partner. In addition, proximity ligation and coimmunoprecipitation assays showed physical interactions between FRA-1, c-JUN, JUND, and TEAD1. Accordingly, YAP/TAZ/TEAD synergized with AP-1 in the control of oncogenic growth both in vitro and in orthotopic xenografts. Remarkably, the induction of mammospheres, reflecting the fraction of CSCs/TICs (Tumor-Initiating-Cells) in response to the ectopic expression of a constitutively active TAZ (TAZ-S89A) derivative in MCF10A cells, was antagonized by FRA-1 knockdown [71].

These findings indicate the functionally relevant cooperation between the chromatin-bound FRA-1/AP-1 and YAP/TAZ/TEAD nuclear factors through large numbers of composite regulatory elements in TNBC cells. Moreover, recent analyses of ZEB1 genomic binding sites in breast cancer show that ZEB1 binding sites extensively overlap with both AP-1 (33% ZEB1 overlap with c-JUN) and YAP binding sites. In addition, ZEB1 physically interacts with c-JUN, FRA-1, and YAP. The EMT transcriptional programs are epitomized by the transactivation of mesenchymal genes, associated with the repression of epithelial gene subsets, prototypically represented by *CDH1* (encoding for E-cadherin). An important difference has emerged from the comparison of the genomic binding sites of the subsets of ZEB1-induced genes with respect to ZEB1-repressed genes. While the ZEB/YAP/AP-1 elements coincide with the positively regulated genomic elements, the ZEB1-only ChIP-seq peaks are associated with the negatively regulated genomic elements. Therefore, FRA-1, being the major c-JUN heterodimeric partner in TNBC cells, critically contributes to EMT in cooperation with YAP by selecting the target genes which are induced rather than repressed by ZEB1 [6].

The proposed model on the role of the FRA-1/AP-dimes in the selection of ZEB1-induced vs. ZEB1-repressed genes is represented in Figure 4.

### 3.2. The FRA-1 Target Genes in the Paracrine Control of EMT and Extracellular Proteolysis

In addition to the above-described core EMT-TFs, a variety of FRA-1-regulated genes, encoding for cytokines, extracellular proteases, and receptors, are involved in the EMT mechanisms in breast cancer.

EMT depends on multiple autocrine and paracrine interactions between cancer cells and non-neoplastic cell populations, such as CAFs, TAMs, mesenchymal stem cells (MSCs), and myeloid cells, recruited to the Tumor Microenvironment (TME). In this context, interleukin-6 (IL6) and TGF-beta likely represent the best-characterized mediators of EMT, implicated in the mechanisms triggering the mesenchymal transitions at the tumor–stroma interface [72] (Figure 5).

FRA-1 was characterized as a transcriptional regulator of *IL6* in a pioneering study on the mechanisms of AP-1-1 and NF-kB- mediated induction of the *IL6* promoter in invasive BCCs [73]. Moreover, the PKC-theta-mediated FRA-1 phosphorylation, associated with the invasive front of mammary tumors, results in increased FRA-1 transcriptional activity leading to *IL6* induction in BBCs [48].

Interestingly, FRA-1 controls the *IL6* transcription not only in neoplastic cells but also in macrophages recruited by paracrine signals to the tumor microenvironment, in which the TAMs represent a major source of IL6. Conditioned media from BBCs (4T1) induce FRA-1 accumulation in macrophages (RAW264.7), in which FRA-1 transactivates the *IL6* promoter [74,75]. In addition to its major role as an EMT inducer in breast adenocarcinoma and other epithelial tumors, IL-6 is critically involved in macrophage polarization. In breast tumors, the FRA-1-dependent secretion of IL-6 promotes the generation of M2d macrophages [74] through mechanisms negatively controlled by the miRNA-mediated downregulation of FRA-1 expression [37].

In the EpH4-Fra-1 murine cell system, TGF-beta secretion was increased in response to Fra-1-overexpression, and a TGFBR1 inhibitor partially restored the expression of the epithelial markers downregulated by the ectopic Fra-1 oncoprotein. Accordingly, Fra-1 directly binds and transactivates the *Tgfb1* promoter [66]. The signaling pathways and mechanisms of cooperation between TAMs and BCCs are summarized in Figure 5.

In addition to the above-described EMT-TFs and cytokines, the FRA-1/AP-1 complexes drive the transcription of cellular components critically implicated in Extra-Cellular Matrix (ECM) degradation.

Multiple Matrix Metalloproteinases (MMPs), along with the serine protease urokinase-type Plasminogen Activator (uPA), encoded by *PLAU*, play major roles in the ECM degradation, responsible for the EMT-associated invasiveness of neoplastic cells. The urokinase-dependent extracellular proteolysis is finely controlled by interactions with other components of the plasminogen activation system (uPAR and PAI-1). Interestingly, the three cognate transcripts (encoded by *PLAU*, *PLAUR*, and *SERPINE1*) are upregulated in response to FRA-1 overexpression in a murine breast adenocarcinoma cell system [56].

The FRA-1-dependent transcriptional control of uPA expression in aggressive breast cancer is mediated by complex interactions between the *PLAU* enhancer and promoter regions [76]. Interestingly, the knockdown of PLAU along with MMP9, which is also FRA-1-regulated in breast cancer [58,77], restores the cell–cell adhesion and inhibits the expression of EMT-associated genes in BCCs [78].

Expression profiling of invasive breast tumors shows that *PLAUR* highly correlates with *FOSL1*. In a cohort (1093 patients) of the TCGA database [79], *FOSL1* is among the top five genes coexpressed with *PLAUR*. While other EMT-related genes (e.g., *SNAI1* and *VIM*) are highly coexpressed, *ESR1* (Estrogen Receptor alpha) is one of the top anticorrelated genes, in agreement with the *FOSL1*-associated mesenchymal features of triple-negative (and basal-like) breast cancers.

FRA-1 controls the *PLAUR* promoter in response to the oncogenic signals [80]. Given the role of uPAR in EMT induction in the BCCs [81], *PLAUR* represents a functionally relevant FRA-1/AP-1 downstream effector. The functional cross-talk between FRA-1 and uPAR in driving cell polarization, motility, and invasiveness has been elucidated in human colon carcinoma cell lines [82]. In response to ERK-dependent signaling, FRA-1 inactivates β1-integrin and downregulates RHOA activity, thus enabling the activation of RAC by uPAR, necessary to form polarized lamellipodia extensions. The two ERK-dependent events do not act independently but rather in cooperation since the expression of *PLAUR* depends at least partly on the FRA-1 activity. Moreover, FRA-1 is posttranscriptionally regulated by the integrin-uPAR signaling pathway in BCCs. The vitronectin-induced activation of uPAR triggers the SRC-FAK-MEK-ERK2 pathway is responsible for FRA-1 phosphorylation and stabilization, thus contributing to breast cancer cell invasiveness [83].

Despite the apparent paradox concerning the role of PAI-1 as an inhibitor of the urokinase proteolytic activity, the expression of the cognate gene (*SERPINE1*) strongly correlates with breast cancer metastasis and, in combination with uPA, represents a validated prognostic biomarker [84]. *SERPINE1* is a major TGF-beta-SMAD target gene in breast cancer, and the specific complexes formed by SMAD2/3 and FRA-1/AP-1 dimers are implicated in the TGF-beta-mediated induction of the *SERPINE1* and other promoters, such as *MMP10*, in preneoplastic BCCs [85].

Therefore, the cooperation between FRA-1 and the TGF-beta pathway includes both the FRA-1/AP-1-mediated *TGFB1* transcriptional induction [66] and the interaction between the SMAD2/3 transducers and FRA-1 on genomic target sites [85].

Along with CAFs, TAMs, immune cells, and mesenchymal stem cells, endothelial cells participate in the paracrine interactions in the context of the tumor microenvironment. Interestingly, the metastatic dissemination of TNBC cells depends on the crosstalk between cancer cells and endothelial cells. The tumor cells secrete the FRA-1-regulated PAI-1, which stimulates the expression of the chemokine CCL5 from endothelial cells. In their turn, CCL5 acts on TNBC cells through a paracrine mechanism to stimulate migration, invasion, and metastasis [86].

In addition to direct transactivation, FRA-1 induces the components of the plasminogen activation system components also by indirect pathways mediated by FRA-1-controlled factors. Interestingly, secretome analyses show that the extracellular accumulation of uPA, PAI-1, and uPAR tightly depends on the expression of the HMGA1 [87], which is transcriptionally controlled by the FRA-1/AP-1 [88].

HMGA1 is a non-histone chromatin component that facilitates the recruitment of transcription factors and chromatin modifiers to modulate gene expression. HMGA1, which is overexpressed in ESCs and downregulated in differentiated tissues, accumulates in virtually all tumor types [89]. Seminal work, based on AP-1 inhibition by a c-JUN dominant-negative derivative, showed that *HMGA1* was an essential AP-1 target gene in the tumor-promoter-induced transformation [90]. In basal-like breast cancer, *HMGA1* is a member of a nine-genes ES-like signature [91]. HMGA1 participates in TNBC metastasis mechanisms by inducing EMT and expression of aggressiveness- and stemness-related factors [92], along with VEGF-mediated angiogenesis [90,93].

### 3.3. The FRA-1-Regulated Genes as Therapeutic Targets

Differing from the poorly druggable transcription factors, such as FRA-1, several members of the FRA-1/AP-1 transcriptome encode for cellular components, such as receptors, which can be pharmacologically targeted, to inhibit the FRA-1-driven EMT.

The tyrosine kinase receptor AXL is tightly coexpressed with FRA-1 in a variety of invasive cancer cell lines, including muscle-invasive bladder carcinoma cells, in which *AXL* was originally identified as an FRA-1 transcriptional target involved in the control of cancer cell motility [94].

AXL contributes to the invasiveness, metastatic dissemination stemness, and chemoresistance of TNBC cells, in which AXL is autocrinally activated by its own ligand (Gas6). Both the receptor and the ligand are often coexpressed in human breast cancer, in which AXL represents a strong negative prognostic factor and a downstream effector of EMT-TFs, as shown in preneoplastic cells ectopically expressing SLUG and SNAIL [95]. In the same cell system (MCF10A), AXL overexpression, in turn, activates a positive feedback loop mechanism by inducing both SLUG and SNAIL and regulating self-renewing of breast CSCs [96].

*AXL* and *FOSL1* are connected by a therapeutically relevant positive feedback loop: in addition to being transcriptionally regulated by FRA-1, the AXL tyrosine kinase is an upstream regulator of *FOSL1*. AXL can be targeted by an antibody interfering with the natural ligand GAS6. In TNBC cell xenografts or patient-derived xenografts (PDXs), the therapeutic antibody inhibits the GAS6-mediated induction of cell migration and invasion and FRA-1 expression, along with the core EMT-TFs (ZEB1/2, TWIST, SNAIL and SLUG) and vimentin.

The downregulation of AXL using MP470 (Amuvatinib) reverts the EMT triggered by TGF-beta and TNF-alpha in preneoplastic cells along with decreasing the breast CSCs self-renewal and chemoresistance [96,97]. In addition to Amuvatinib, which also targets other RTKs (PDGFR, c-KIT, MET), more selective orally bioavailable AXL inhibitors have been recently characterized as promising therapeutic tools against metastatic breast cancer [98].

Other promising therapeutic targets have been revealed by functional studies on the FRA-1 transcriptome. High-throughput synthetic lethality screens, aimed at identifying drugs selectively killing the metastatic FRA-1–overexpressing, with but not the FRA-1-depleted MDA-MB-231 BCCs, resulted in the identification of an FRA-1 target gene (*ADORA2B*) encoding a pharmacologically tractable adenosine receptor. Among the adenosine receptor antagonists, the highly tolerable bronchodilator theophylline strongly synergized with docetaxel in inhibiting the metastatic activity of BCCs [7].

Although the ADORA2B knockdown recapitulated the inhibitory effect of theophylline on metastatic dissemination, other mechanisms should be considered. In addition to antagonizing the adenosine receptors, the theophylline-mediated inhibition of phosphodiesterase induces cAMP-PKA signaling. This pathway is known to induce MET (Mesenchymal-to-Epithelial Transition) in NAMEC and inhibit the TICs fraction in the Ras-transformed derivative [99].

Since theophylline strongly inhibited the tumor-initiating ability of NAMEC-Ras cells [97], and given the synergism between theophylline and docetaxel in MDA-MB-231 cells [7], it will be important to investigate the possible ADORA2B-independent mechanisms of restoration of epithelial features. For example, as shown for other PKA agonists, such as forskolin, which improves the sensitivity to doxorubicin by inhibiting the ERK activity in MDA-MB-231 cells [100], the reversion of EMT in the same cell system might be consequent to the theophylline-induced downregulation of ERK activity and FRA-1 expression.

### 3.4. The Role of FRA-1-Repressed Genes in the Control of EMT

Analysis of the FRA-1/Fra-1 transcriptomes in both human and mouse cell systems indicates the functional relevance of several genes negatively controlled by FRA-1/Fra-1. In the mouse Eph4 cells, the gene sets and pathways implicated in cell junction organization and tight junctions were downregulated in response to the ectopically expressed Fra-1 [66]. In a human cell system (BT549), twenty proliferation-repressive genes were upregulated in response to the c-JUN/FRA-1 combined knockdown. In addition to well-known EMT inhibitors, such as *CDH1* (encoding E-cadherin), the downregulated subset also included genes with tumor suppressor activity and poorly characterized roles in EMT. Among the FRA-1-downregulated genes, *CLCA2*, encoding for the Chloride Channel Accessory 2 protein, is targeted by FRA-1 and c-JUN, interacting with the *CLCA2* third intron. CLCA2 was previously characterized as a p53-induced inhibitor of cell proliferation [101] and, more recently, as a prognostically relevant inhibitor of EMT in TNBC cells. It was proposed that by inhibiting the chloride current, the CLCA2 downregulation contributes to the increased pHi and metabolic changes associated with cancer cell invasiveness [102]. According to more recent findings, CLCA2 is involved in the maintenance of the junctional anchoring and epithelial state by colocalizing at the cell-cell junctions and interacting with EVA1 and ZO-1. Moreover, in the membrane, CLCA2 colocalizes with E-cadherin and interacts with beta-catenin regulating homophilic cell–cell interactions while inhibiting the beta-catenin cytosolic signaling and downregulating the EMT-inducing beta-catenin target genes [103].

The FRA-1-downregulated genes also include indirect targets, posttranscriptionally controlled by negative regulators induced by FRA-1 the gene products. The miRNA-mediated regulatory circuits are highly relevant in EMT, as shown by the double-negative feedback loops formed by the EMT suppressor miRNAs (miR-200, miR-34, and miR-15/16 family members) and the core EMT-TFs. In BCCs, FRA-1 downregulates the transcriptional repressor TRPS1 through a miRNA-mediated mechanism [69]. *TRPS1*, which anticorrelates with *FOSL1* in aggressive breast carcinoma (TCGA expression Atlas [79]), encodes for a GATA-type zinc-finger transcription repressor playing cell context-dependent roles. In mammary gland development, *TRPS1* is essential for proliferation and lactogenic differentiation. In breast cancer, the loss of TRPS1 along with loss/inactivation of E-cadherin results in increased proliferation of mammary organoids and accelerated tumorigenesis in mouse models [104]. The FRA-1 effect on TRPS1 is mediated by the miR-221/222 oncomiRs, which are overexpressed in basal-like breast cancers. In response to the RTK/RAS/RAF/ERK pathway, FRA-1 binding to the miR-221/222 promoter region induces the expression of both miRNAs, which, in turn, target the *TRPS1* transcript and downregulate the protein product. Among the downstream genes subjected to TRPS1-mediated repression, ZEB2 is strongly implicated in the effect of TRPS1 downregulation on the EMT induction [69]. Therefore, the FRA-1-miR-221/222-TRPS1-ZEB2 axis further reinforces the regulatory links between Fra-1 and ZEB2 in breast cancer.

Other oncomiRs are likely implicated in the FRA-1-mediated EMT mechanisms. The well-characterized onco-miRNA miR-21 is overexpressed in almost every cancer cell type. Differently from miR-221/222, miR-21 expression does not correlate with specific subtypes, such as the basal-like. However, the miR-21-mediated inhibition of the anti-metastatic gene (*LZTFL1*) is required for the in vitro invasiveness and in vivo metastasis along with the EMT markers in BCCs [105]. Since FRA-1 is a transcriptional regulator of miR-21, at least in some cell contexts, miR-21 might participate in the FRA-1-mediated EMT induction.

### 3.5. FRA-1 Effects on the Architecture of Target Promoters

The characterization of FRA-1-binding genomic elements has highlighted several mechanisms of FRA1-dependent chromatin modifications and transactivation of target genes.

Dissection of the *PLAU* regulatory region shows that FRA-1 binding to the proximal (−1.9 kb) enhancer mediates the recruitment of the p300 HAT, which is associated with the transcription of multiple RNA species along with the bona fide *PLAU* mRNA precursor. These short unstable RNAs, transcribed bidirectionally across the enhancer regions upstream to the *PLAU* TSS, likely act as enhancer RNA molecules (eRNAs), regulating the epigenetic state of the chromatin [106]. In addition, a relatively stable transcript, extending from the −1.9 kb enhancer encompassing the *PLAU* mRNA from the TSS to the 3′UTR, is expressed in multiple basal-like cancer cell lines, although at low level with respect to the productive *PLAU* mRNA. Interestingly, FRA-1 positively controls the *PLAU* mRNA but inhibits the accumulation of the longer transcript initiating in the 5′ flanking region [76]. The functional significance of these findings awaits further investigations.

While the *PLAU* transcription mainly depends on the tracking by RNA polymerase of the region upstream to the *PLAU* locus, the ZEB2 transcriptional regulation points to the role of FRA-1 in the control of long-range chromatin interactions. The TNF-alpha-mediated induction of EMT in TNBC cells (BT549) is associated with the expression of two alternatively spliced ZEB2 mRNA isoforms transcribed from two distinct promoters located 2.4kb apart. FRA-1/c-JUN dimers are recruited on the distal promoter and mediate the looping of both promoters, which is essential for *ZEB2* transcriptional induction.

The FOS family member FRA-2 is often co-expressed with FRA-1 in TNBC cells, and early findings suggested that FRA-2 could cooperate with FRA-1 in the control of breast cancer cell motility and invasiveness [107]. These findings raise the question of FRA-2’s contribution to the regulation of FRA-1 target genes. Interestingly, the AP-1-mediated control of *HMGA1* transcription shows that, although both FRA-1 and FRA-2 bind to the last two introns of the gene, only FRA-1 is required for *HMGA1* transcription. Chromatin conformation analyses show that the distal FRA-1 (and FRA-2) binding region interacts with the *HMGA1* promoter. Surprisingly, however, at variance with ZEB2, FRA-1 is not required for the DNA looping. Moreover, as in the case of *PLAU*, FRA-1 expression is required for the recruitment of p300 to the enhancer region, but quite surprisingly, p300 (as well as CBP) is not required for *HMGA1* transcription in MDA-MB-231 cells. In summary, the FRA-1-independent interaction between the enhancer and promoter regions allows the enhancer-bound FRA-1 to drive the recruitment of RNA polymerase (without affecting the Pol II CTD P-Ser5/P-Ser2 ratio) on the *HMGA1* promoter [88]. Altogether, these studies suggest that FRA-1 controls its targets by gene-specific mechanisms.

A general picture has emerged from the in-depth analysis of FRA-1 and FRA-2 transcriptomes and genomic distribution, with the associated epigenomic modifications and long-range interactions. In agreement with its biological functions, FRA-1 regulates many more genes and exerts stronger transcriptional effects with respect to FRA-2. Genomic distribution, along with associated histone modifications, RNA Polymerase, and CBP/p300 recruitment and chromatin accessibility, shows that FRA-1 and FRA-2 prevalently bind to enhancers rather than promoter regions. FRA-1 controls the recruitment of CBP/p300 both positively and negatively, depending on the target. Moreover, as observed for *HMGA1*, FRA-1 exerts limited effects on the chromatin structure (DNA looping) of selected target genes, thus raising the question of the role of Fra-1 interactions with other transcription factors and coactivators within the context of 3D enhancer hubs [9].

Other lines of research stemming from the mass spectrometry-based identification of chromatin-bound FRA-1 interaction partners will contribute to shedding light on the mechanisms of FRA-1-mediated transactivation regarding EMT and therapeutic applications. The RNA helicase DDX (DEAD-box)5/p68, identified as the most enriched FRA-1-binding protein among 118 interactors in the chromatin of TNBC cells (BT549), shares with FRA-1 the majority (62%) of its genomic binding sites. DDX5 is essential for the transcriptional activation of FRA-1 target genes, and both positively and negatively regulated DDX5-dependent gene sets exhibit prognostic value in breast cancer patients. Accordingly, DDX5 is overexpressed in basal-like tumors, and DDX5 levels are predictive of worse outcomes in ER- but not ER+ breast cancers [108].

Recent findings, originating from seminal work in colorectal cancer cells [109], show that DDX5 promotes EMT by inducing the expression of the PDGF receptor in TNBC cells [110]. Therefore, given the role of the PDGF autocrine signaling in the maintenance of the FRA-1-driven EMT and cancer cell stemness [19], DDX5 might collaborate with FRA-1 by both direct mechanisms, mediated by the DDX5-FRA-1 interaction on chromatin [108], and indirect mechanisms, by sustaining the expression of PDGFR in TNBC cells [110]. Therefore, in addition to being posttranscriptionally regulated by RBPs (IGF2BP1) interacting with the *FOSL1* transcript, FRA-1 also cooperates with other RBPs, such as the RNA helicase DDX5, in invasive breast cancer.

The same proteomics analysis [108] points to PARP1 as a therapeutically promising FRA-1 interactor. The interaction with PARP1 results in FRA-1 PARylation (Poly-ADP-Ribosylation), which can be inhibited by treatment with olaparib. The PARP inhibitor upregulates the FRA-1 protein and mRNA expression levels by mechanisms likely dependent on the AP-1-mediated *FOSL1* transcriptional autoregulation. On the other hand, FRA-1 knockdown potentiates the proapoptotic effects of olaparib in TNBC cells. Transcriptomic analyses showed that a large fraction of the olaparib-regulated genes is induced in an AP-1-dependent manner. Remarkably, the EMT signaling pathway is one of the most enriched pathways in response to olaparib treatment, in agreement with the evidence that many of the olaparib-induced genes are direct FRA-1 targets [111]. These findings are translationally relevant considering that the FRA-1 inhibition could antagonize the mesenchymal transition elicited by the PARP inhibitor in olaparib naïve BRCA2/p53-mutant mammary tumors in mice [112].

### 3.6. Diagnostic and Prognostic Significance of the FRA-1 Oncoprotein and FRA-1-Derived Signatures in TNBC

The clinicopathological relevance of FRA-1 overexpression was originally suggested by immunohistochemical analyses of breast cancer cell lines and tumors, which showed the prognostically significant inverse correlation between the expression of FRA-1, accumulating in less differentiated cancers, and FOSB, expressed in normal mammary glands and well-differentiated tumors [113]. Subsequent analyses of breast cancer-associated AP-1 compositional changes showed the strong inverse correlation of FRA-1 vs. ER and PR expression, along with the FRA-1 accumulation in triple-negative compared to luminal carcinomas [114].

At the mRNA level, analyses of microarray data sets representing large cohorts of breast cancer patients showed that FRA-1 mRNA level inversely correlated with DMFS (Distant Metastasis-Free Survival), while c-FOS expression exhibited the opposite correlation [66].

The clinicopathological correlations were further reinforced by analyzing the FRA-1 transcriptome, representing a surrogate readout of FRA-1 activity in highly metastatic lines derived from the phenotypically mesenchymal MDA-MB-231 cells. A prognostically relevant subset of FRA-1 targets, further restricted to a 183-genes Fra-1 classifier, was defined by bioinformatic analysis. Remarkably, the FRA-1 classifier performed better than other available prognostic signatures in predicting the outcome (time to distant metastases or relapse) in TNBC patients [7]. The predictive value of the FRA-1 classifier was further delineated through a functional approach. A nine-genes subset was generated by investigating the effects on primary and/or metastatic growth of inhibition of 31 prognostically relevant FRA-1-regulated genes. Interestingly, within this subset, *EZH2* (Enhancer of Zeste Homolog 2, the enzymatic component of the Polycomb Repressive Complex 2 PRC2 catalyzing the H3K27 trimethylation) represents a further functional axis linking FRA-1, lncRNAs, and EMT-TFs. An important link between EZH2 and EMT is represented by the SNAIL-mediated recruitment of EZH2 to specific genomic sites via interactions through the lncRNA HOTAIR (for HOX Transcript Antisense Intergenic RNA) during TGF-beta-induced EMT in human hepatocytes [115]. Moreover, in esophageal cancer cells, EZH2 reinforces the EMT features and drug resistance by upregulating the ZEB1 expression through the interaction with the lncRNA LINC00152 [116]. Remarkably, the nine-genes subset retains prognostic significance in both ER-positive and ER-negative breast cancer subtypes [117].

More recently, a four-gene prognostic signature predictive of poor overall survival has been characterized in basal B BCCs, in which constitutive FRA-1 phosphorylation is sustained by the FAK-SRC-MEK-ERK pathway in response to the integrin uPAR signaling. Interestingly, the signature includes the uPAR-coding gene, *PLAUR*, along with two uPAR ligands (*PLAU* and *VTN*) and *FOSL1* itself [83].

Finally, powerful prognostic correlations originate from the recent analyses of cooperation mechanisms between *FOSL1*, *ZEB1*, and *YAP* in breast cancer. *FOSL1* is highly coexpressed with *ZEB1*, and the prognostic value (relapse-free survival) of *FOSL1*+ *ZEB1* expression is higher than that of *ZEB1* alone. Accordingly, the high expression of a subset of eight common ZEB1/YAP/AP-1 target genes correlates with lower relapse-free and DMFS (Distant Metastasis-Free Survival) rates [6].

## 4. Open Questions and Novel Perspectives

### 4.1. The Role of Long Non-Coding RNAs as Downstream Effectors of FRA-1

In addition to the miRNAs playing well-characterized roles as posttranscriptional regulators of EMT, thousands of lncRNAs affect cancer progression through many different molecular mechanisms.

Several lncRNAs involved in EMT and breast cancer metastasis have been characterized. Representative examples include the lncRNA H19 and the TGF-beta-induced lncATB, which affect EMT by acting as sponges for miR-200 and let-7 family members [118,119], while the lncRNA MALAT1 acts by sequestering the TEAD transcription factor and preventing its interaction with YAP [120].

Many potential EMT-related lncRNAs are expressed in TNBC (reviewed in [121]), and many lncRNAs are significantly coexpressed with FRA-1 in TNBC (Breast Invasive Carcinoma dataset, TCGA Pan-Cancer Atlas, 1082 samples [79]). These observations prompt to investigate the role of lncRNAs transcriptionally regulated by FRA-1/AP-1 in concert with the above-summarized protein-coding genes and miRNA precursors in the multi-layer control of EMT in BCCs.

### 4.2. The In Vivo Analysis of Fra-1 Functions in Mouse Models of TNBC

Despite a large amount of evidence from in vitro cell systems and to the best of our knowledge, the Fra-1 function has not been investigated in mouse models of invasive breast cancer. However, several genetically modified mouse models are amenable to in vivo analyses of Fra-1 function in EMT and metastasis mechanisms.

The role of Fra-1 in EMT has been highlighted in a conditional knockout model in which spontaneous breast tumors develop in the absence of the *Scrib* gene product, characterized as a regulator of cell polarity. *Scrib* ablation in the mammary gland perturbs the luminal differentiation causing disruption of the epithelial cells’ polarity associated with the expansion of poorly differentiated intraluminal cells [122]. Scrib was originally described as a tumor suppressor in Drosophila, and cooperation between *Scrib* loss and *Myc* expression is involved in the mammary tumorigenesis [123]. *Scrib* knockdown also results in increased MAPK signaling and invasiveness in response to H-Ras [124]. Remarkably, in the spontaneous tumors arising in *Scrib* knockout mice, the Mek/MAPK-Fra-1 pathway is constitutively active, and Fra-1 accumulation correlates with the disruption of epithelial integrity and gain of mesenchymal features comparable to human basal-like breast cancer [122].

Important results on the in vivo cooperation between a breast cancer-associated EMT-TF (Twist1) and the Ras oncoprotein have been obtained in a mouse model, in which the mammary gland-specific expression of the Cre recombinase is driven by the WAP (Whey Acidic Protein) promoter. The multifocal breast carcinomas developing in WAP-Cre; K-rasG12D; Twist1 transgenic females exhibit features of mesenchymal transdifferentiation and gene expression profiles associated with the claudin-low subtype [125]. By similar approaches, the *Fosl1* transgenic expression could shed light on the in vivo cooperation between Fra-1 and K-ras in breast cancer initiation and progression.

The in vivo roles of Fra-1 in invasive breast cancer can be investigated in the available models of TNBC. The mammary gland-specific overexpression of Met associated with the loss of p53 results in highly penetrant tumors with the features of the claudin-low subtype. These tumors exhibit mesenchymal features correlating with the overexpression of the six EMT-TFs, and gene expression profiling shows the expected *Fosl1* overexpression [126].

A distinct mouse model developing triple-negative claudin-low carcinomas has been generated by transgenic expression of the prolactin gene (*Prl*) combined with *Trp53* ablation. Interestingly, along with the mesenchymal features and aggressive behavior, the *Prl*/*p53*-/- breast tumors also exhibit increased Fra-1 and c-Jun protein levels [127].

The above-described TNBC murine models prompt to investigate the consequences of the loss of Fra-1 in genetic crosses carrying the floxed *Fosl1* allele via the same approach adopted for validating the in vivo role of Fra-1 in K-Ras-induced lung tumorigenesis. The lack of Fra-1 inhibits lung tumor progression and increases survival in a murine system, in which the development of tumors with histopathological features of human LAC (Lung Adenocarcinoma) depends on the tissue-specific induction of mutant K-Ras combined with *Trp53* deletion [128]. Similarly, the effect of *Fosl1* ablation can be analyzed in breast cancer-prone genetic backgrounds exhibiting mammary gland-specific expression of the Cre recombinase (e.g., MMTV-MET; MMTV-Cre; *Trp53*fl/+).

The precise modeling of genetically heterogeneous cancers, such as basal-like/TNBC tumors, requires the analyses of complex combinations of oncogenic lesions. Despite the advantages of the conditional CRISPR-Cas9 systems, many time-consuming crosses can be necessary to generate murine lines exhibiting multiple germline modifications. Promising alternatives are represented by strategies aimed at rapid in vivo generation of somatic mutations. The RCAS-TVA approach is based on Replication-Competent Avian leukosis virus Splice-acceptor (RCAS) vectors, which specifically target the cell populations expressing the avian viral receptor (TVA) [129]. Remarkably, by generating the MMTV-TVA mouse model, the RCAS-TVA system has been exploited for introducing multiple collaborating oncogenes (*Py-MT*, *Wnt-1*, *Neu*) in mammary glands [130]. The same approach, in combination with CRISPR-Cas9, allows the targeted inactivation of tumor suppressor genes and the editing of proto-oncogenes. The GFAP-TVA-dependent glial cell-specific *Nf1* (neurofibromin) inactivation, along with K-RasG12V expression, triggers the growth of the most aggressive glioblastoma subtype (MES), exhibiting mesenchymal features. Interestingly, in the same system, in which Fra-1 overexpression was specifically associated with the MES subtype, *Fosl1* is essential for the PN (ProNeural) to MES (MESenchymal) transition, which can be regarded as an EMT-like process, associated with tumor progression and aggressiveness of human glioblastoma [131].

### 4.3. FRA-1 in Tumor Microenvironment

Among the major consequences of EMT, phenotypical mesenchymal cancer cells exhibit increased resistance to anti-tumor immune attacks through the dysregulated expression of immune checkpoint inhibitors, cytokines, and chemokines. The paracrine signals mediate the recruitment of various inflammatory cell populations, including TAMs, to the tumor stroma [132].

The pro-oncogenic role of TAMs strongly depends upon functional reprogramming (polarization) from the M1 (antitumor) to the M2 (immunosuppressive) phenotype. In in vitro systems, invasive BCCs trigger the FRA-1 expression in cocultured macrophages. In both in vitro and in vivo generated TAMs, Fra-1 drives the differentiation toward the tumor-promoting M2d phenotype through mechanisms dependent on autocrine pathways triggered by the transactivation of the interleukin-6 promoter [37,74].

Interestingly, FRA-1 is also involved in macrophage function in non-neoplastic diseases, such as rheumatoid arthritis, in which FRA-1 promotes the macrophages’ proinflammatory reprogramming. Joint inflammation is ameliorated in arthritic mice lacking FRA-1 expression in myeloid cells (FRA-1ΔMx) or in less heterogeneous cell populations (FRA-1ΔLys), devoid of FRA-1 expression in neutrophils, monocytes, and macrophages. The intercrosses between these strains and the above-summarized models of breast cancer will make it possible to investigate the effect of the myeloid/macrophage-specific FRA-1 ablation on the EMT parameters, metastatic dissemination and response to immune checkpoint inhibitors [133].

The seminal findings on the *Fosl1* requirement for placental vascularization [134], along with the *FOSL1* role in endothelial cell migration and assembly of capillaries [135], suggested the Fra-1 involvement in tumor angiogenesis. The FRA-1-dependent control of VEGFA expression in invasive BCCs [58] highlighted the FRA-1 participation in the VEGF-dependent tumor angiogenesis, which, in turn, contributes to the EMT-induced stemness resulting in the increased tumorigenicity of breast CSCs [136]. Among the available endothelial-specific Cre mouse models [137], the lines expressing the Cre recombinase in sprouting angiogenic cells allow the endothelial-specific ablation of *Fosl1*. Genetic crosses with breast cancer mouse models will elucidate the role of endothelial Fra-1 on tumor-induced angiogenesis and mesenchymal transition mechanisms.

### 4.4. FRA-1 in the Control of the Hybrid E/M (Partial EMT) Phenotypes

The EMT does not represent a strictly binary switch since cancer cells can reside in multiple intermediate states between the fully epithelial and fully mesenchymal phenotypes, as shown in a variety of human tumors and mouse models [138]. In TNBC mesenchymal cells, the prognostically relevant cell surface biomarker CD104/ITGB4 allows the identification of phenotypically intermediate cellular subpopulations exhibiting partial EMT features. The quasi-mesenchymal cell subpopulations harbor the strongest tumor-initiating capacity, reflecting the higher fraction of CSCs, with respect to the fully mesenchymal cell population [139].

The functional interplay between distinct EMT-TFs in control of the hybrid (E/M) vs. fully mesenchymal (M) state has been recently characterized [140]. While the highly tumorigenic hybrid E/M state is mainly driven by SNAIL and canonical WNT signaling, the poorly tumorigenic fully mesenchymal state depends on the constitutive ZEB1 expression, associated with noncanonical (WNT5a-driven) WNT signaling [141].

As discussed above, the SNAIL- and TWIST-mediated induction of *FOSL1* critically contributes to the transcriptional programs supporting the CSC population [59]. In addition to the maximal coexpression with *SNAI1*, representing one of the top five coexpressed genes, in TNBC clinical specimens, *FOSL1* expression strongly correlates with the hybrid (E/M) marker ITGB4, identifying the more aggressive subtypes of mesenchymal carcinoma cells [139]. According to the same dataset (Breast Invasive Carcinoma, TCGA PanCancer Atlas, 1082 samples), ITGB4, previously characterized as a negative prognosticator associated with the basal-like subtype [142], is the beta-family integrin best correlating with *FOSL1* expression level. Along with similar results in colorectal cancer [143], these data suggest that FRA-1 might contribute to the transactivation of the *ITGB4* promoter in invasive breast cancer, as previously shown in undifferentiated basal keratinocytes [144].

In summary, the EMT-associated phenotypic plasticity prompts the investigation of FRA-1 expression and function in the hybrid E/M subpopulation. Given the heterogeneity of cell populations sorted by cytofluorimetry, single-cell RNA-seq analyses are required for elucidating the FRA-1 regulatory networks in the individual subpopulations reflecting distinct transition trajectories, as recently investigated in multiple cancer types [138,145].

FRA-1 is overexpressed in several TNBC cell lines derived from metastatic tumors exhibiting full EMT features. However, because of the loss of plasticity in in vitro propagated cells remaining “frozen” in the fully mesenchymal state, cancer cell lines are poorly suitable for studying the dynamics of FRA-1 through multiple EMT states. Along with single-cell analyses, full elucidation of the transition states associated with distinct neoplastic hallmarks requires in vivo studies in mouse models of invasive breast cancer, in which metastasizing cells can interact with different tissue microenvironments, during multiple steps of local invasion, systemic dissemination, and secondary colonization.

Recently, the in vivo visualization of partial EMT cell states has been exploited by genetic tracing of the activity of genes transiently expressed during different stages of EMT in primary tumors and lung metastases in the MMTV-PyMT mouse model. The fate mapping showed that N-cadherin was activated and functionally required for metastases, while vimentin expression, although associated with mesenchymal features in primary tumors, was not significantly activated or functionally required for lung metastases [146].

By a similar approach in the same model of metastatic breast cancer (MMTV-PyMT), the relative contribution to metastatic colonization of partial vs. full EMT has been shown by in vivo imaging, along with flow cytometry and expression analyses of small cell subsets (5-cell RNA-seq). The results confirm that partial but not full EMT cells are required for metastasis formation, while both partial and full EMT cells exhibit chemoresistance with respect to the epithelial subpopulations [147].

Most reports on FRA-1 in breast cancer dissemination rely on analyses of lung metastases formed by intravenously injected fully mesenchymal TNBC lines (e.g., MDA-MB-231) subjected to *FOSL1* silencing. These approaches do not allow us to assess the relationship between FRA-1 expression and the EMP (Epithelial-Mesenchymal-Plasticity), which has emerged as a key factor in EMT-dependent metastasis mechanisms. We envisage that the FRA-1 temporal and spatial dynamics associated with distinct (i.e., partial vs. full) EMT states, invasion, and metastasis could be fully elucidated by introducing the *Fosl1* allele in which GFP (or a similar marker) recapitulates the Fra-1 expression pattern in tumor prone models, such as the MMTV-PyMT mouse. The results will show if Fra-1 expression prevails in the phenotypically plastic cell populations exhibiting partial EMT markers or in the fully mesenchymal subpopulations, characterized by decreased collective migration and metastatic ability, associated with the gain of maximal chemoresistance. These data could bear relevant therapeutic implications, considering that if Fra-1 is overexpressed in the fully mesenchymal but not in the quasi-mesenchymal subpopulation, partial suppression of Fra-1 expression and/or activity might increase the fraction of phenotypically plastic cell population, thus resulting in increasing the number of metastatic cells. A similar question has been addressed in the case of the therapeutic inhibition of PRC2. Interestingly, the same drug, which can restore the MHC-I expression and increase the efficacy of paclitaxel chemotherapy in the TNBC [148], also induces the EMP and enhances the quasi-mesenchymal cell fraction and metastatic colonization [149].

### 4.5. CTCs (Circulating Tumor Cells) and PDXs as Tools for In Vivo Studies on FRA-1 in Human Cancer

Studies on in vivo EMT in human specimens are hampered by the difficulty in identifying the phenotypically mesenchymal vs. epithelial cells, both in primary tumors and in metastatic lesions, where colonizing cells undergo the Mesenchymal-to-Epithelial Transition. However, several recently developed methods for the enrichment of the rare CTCs allow the study of the EMT hallmarks in cancer cells isolated from the bloodstream. Multiple studies on human CTCs support the key role of the hybrid E/M cell populations, in agreement with the results from in vitro systems and animal models. Detection of both epithelial and mesenchymal markers, along with partial retainment of intercellular adhesions, in clusters of CTCs from breast cancer patients, favors a mechanism of collective cell migration, associated with higher metastatic potential compared to single CTCs [150,151,152]. These findings prompt to investigate the role and prognostic significance of FRA-1 in CTCs, supported by the results obtained in the TNBC mouse model, in which CTCs isolated from tumor-bearing mice exhibited Fra-1 overexpression associated with distinct morphology with respect to primary tumors [49].

Promising alternatives to genetically modified animals included the orthotopic mouse models based on intraductal implantation of murine TNBC-like cell lines in syngeneic mice, followed by spontaneously developing metastases [153]. In addition to representing the cancer avatars for personalized therapies [154], the PDXs established from the major breast cancer subtypes are amenable to studies on tumor microenvironment and metastatic progression in the clinically heterogeneous TNBCs. Regarding EMT, PDXs have been recently exploited for investigating the role of EMT-TFs and BRD4 in in vivo dissemination of TNBC cells [18]. Accordingly, we envisage that, along with the approaches based on single-cell analyses, the recently available biobanks of PDXs and PDX-derived organoids will represent an important tool for innovative investigations of the FRA-1 roles in human EMT and metastasis mechanisms [155].

### 4.6. FOSL1/FRA-1 as a Target of Therapeutic Intervention in TNBC/Basal-like Breast Cancer

Because of its pivotal role in transcriptional control of EMT and metastasis, the FRA-1 oncoprotein is an ideal therapeutic target. However, because of the largely disordered 3D structure, as most transcription factors, FRA-1 represents a poorly druggable target. In addition, being activated by overexpression rather than activating mutations, CRISPR base editing is not suitable for *FOSL1* therapeutic correction. In a recent review article, based on up-to-date technological advancements, we examined several innovative strategies aimed at the in vivo inhibition of FRA-1 expression and/or activity [54]. We have discussed the pros and cons of the approaches aimed at the CRSPR-mediated *FOSL1* knockout, along with the RNA-based therapeutics for reversible inhibition of *FOSL1* transcription and/or translation. We have also reviewed several recent strategies (PROTACs) aimed at FRA-1 destabilization, along with suicide gene therapies leveraging on the phosphorylation-mediated control of FRA-1 half-life [156,157,158].

Although several innovative approaches for specific targeting of DNA and RNA therapeutic molecules are under investigation in preclinical models, the hurdles concerning cancer cell-specific delivery point to the importance of small molecule inhibitors. While several compounds acting as AP-1 inhibitors have been characterized and tested in clinical trials, the search for FRA-1-specific inhibitors is still in progress [54].

Polypeptide inhibitors include recombinant or synthetic polypeptides acting as bZIP competitors, forming DNA binding-incompetent heterodimers. Interestingly, a 39-aa peptide, specifically inhibiting the DNA binding activity of FRA-1-containing AP-1 dimers, has been identified by exploiting the isCAN computational tool, an effective methodology coupling library design with computational (bZIP Coiled Coil Prediction Algorithm, bCIPA), and intracellular screening. The selected peptide (FRA1W) shows a binding affinity for FRA-1 within a nanomolar range and does not homodimerize or heterodimerize with the JUN-family members [159].

These findings point to the results expected from the application of Artificial Intelligence (AI) based tools, which have revolutionized the field of protein modeling. Based on the AlphaFold AI algorithm developed by DeepMind [160,161], the FRA-1 structure has been modeled. Only the 70–80-aa encompassing the bZIP region exhibits a very high (>90) per-residue confidence score (pLDDT), while most of the protein appears intrinsically unstructured (https://alphafold.ebi.ac.uk/entry/P15407, accessed on 13 December 2022). As for other intrinsically disordered proteins, interactions with dimeric partners and DNA likely induce the folding of the unstructured FRA-1 regions. While the AlphaFold, although considered the gold standard of protein prediction, is limited to single protein structures, RoseTTAFold expands the structure modeling capabilities from monomeric proteins to large protein assemblies [162]. We envisage that the RoseTTaFold method might allow the accurate prediction of FRA-1-containing multiprotein complexes and the design of potent and selective FRA-1 inhibitors [163], leading to the development of novel drugs for the inhibition of the FRA-1/AP-1-controlled transcriptional networks of EMT and metastasis in aggressive breast cancer.

## Figures and Tables

**Figure 1 ijms-24-08307-f001:**
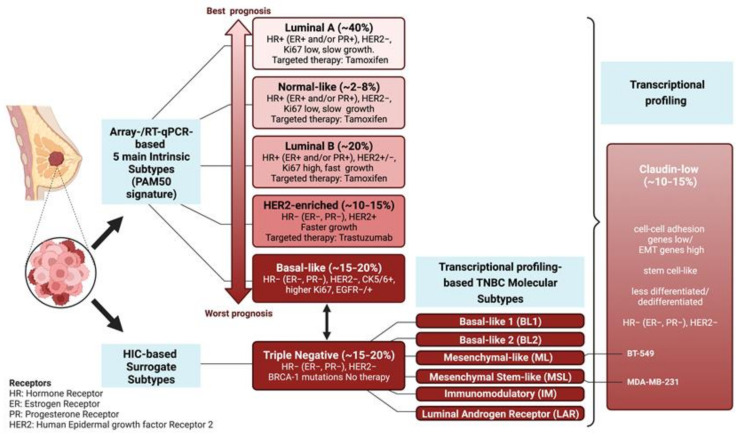
Molecular Classification of Breast Cancer. The five intrinsic breast cancer subtypes (luminal-A, normal-like, luminal-B, HER2-enriched, and basal-like) based on the PAM50 expression signature are shown in the left column [1,2]. Below is the IHC-based classification of the TNBC surrogate subtype: the basal-like subtype is commonly referred to as TNBC, although not all basal-like are triple-negative and vice versa. TNBCs are further characterized by expression profiling into six molecular subtypes (BL1, BL2, ML, MSL, IM, LAR). The vertical bracket on the side of claudin-low subtype indicates that the claudin-low tumors, rather than representing a subfraction of TNBC, can pervade the five intrinsic subtypes. FRA-1 overexpression is associated with the TNBC/Basal-like subtype. *FOSL1* (along with *ZEB1* and *YAP*) overexpression marks the claudin-low subtype, which exhibits EMT-like and stem-like expression signatures, along with high levels of MAPK pathway activation and stromal infiltration. The FRA-1 transcriptomes and cistromes have been characterized in the mesenchymal-like (ML) BT-549 and the mesenchymal-stem-like (MSL) MDA-MB-231 cell lines, both representing the claudin-low subtype [6,7,8,9].

**Figure 3 ijms-24-08307-f003:**
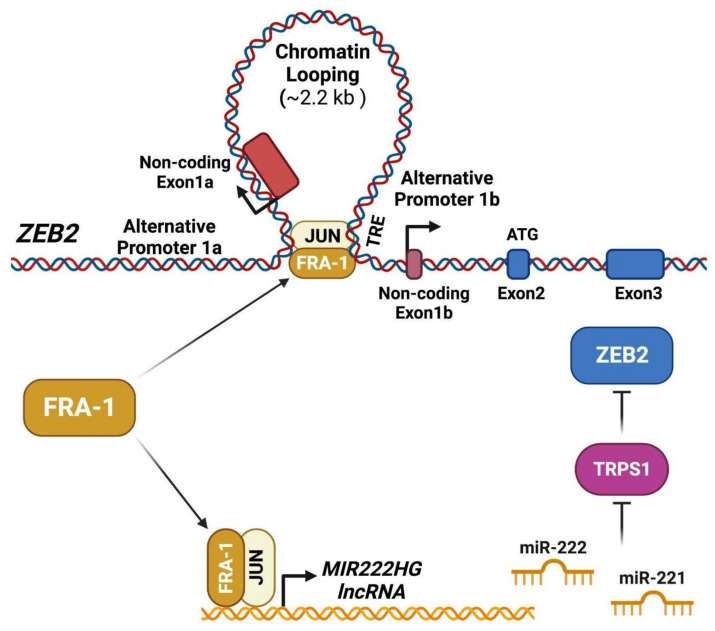
FRA-1/AP-1 controls ZEB2 accumulation and activity by direct and indirect mechanisms. FRA-1/AP-1 drives the transcription from both proximal (1b) and distal (1a) *ZEB2* promoters by mediating DNA looping and long-range interactions [68]. The FRA-1/AP-1 dimers also induce the expression of the co-transcribed miRNAs miR-221 and miR-222 through binding to the transcriptional promoter of the MIR222HG (miR-221/222 cluster Host Gene). The FRA-1-induced miR-221 and miR-222, in turn, downregulate the transcript encoding for the EMT inhibitor TRPS1, acting as a transcriptional repressor of GATA-regulated genes, including *ZEB2*. Therefore, the FRA-1-induced miR-221 and miR-222, by inhibiting the TRPS1 expression, relieve the repression and induce ZEB2 by an indirect mechanism that synergizes with the direct FRA-1-mediated ZEB2 transcriptional induction [69].

**Figure 4 ijms-24-08307-f004:**
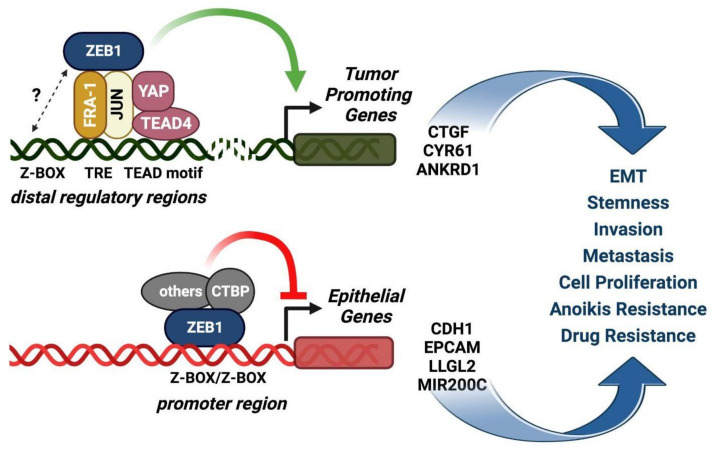
The FRA-1/AP-1 complex acts as a relay for the selection of the ZEB1-induced vs. the ZEB1-repressed target genes. The model (modified from Feldker et al., EMBO J., 2020 [6]) represents the cooperation between FRA-1/AP-1, TEAD4/YAP, and ZEB1 in EMT transcriptional regulation. *Upper drawing*: the extensive overlap of low-affinity monomeric ZEB1 binding sites with AP-1 and YAP motifs, along with physical interactions between FRA-1/c-JUN and ZEB1, have been elucidated by genome-wide analyses in the claudin-low MDA-MB-231 cell line. FRA-1/AP-1 and TEAD4/YAP predominantly recruit ZEB1 to distal regulatory regions through TEAD4-AP-1 DNA binding sites to positively regulate transcription of tumor-promoting genes. *Lower drawing*: the ZEB1-mediated repression of epithelial genes depends on direct ZEB1 binding to bipartite high-affinity Z-boxes, not overlapping with AP-1 and YAP binding sites and preferentially localized in promoter regions of target genes. The complexes interacting with the ZEB-only binding sites inhibit the expression of epithelial target genes (e.g., *CDH1*) by recruiting transcriptional repressors, such as CTBP.

**Figure 5 ijms-24-08307-f005:**
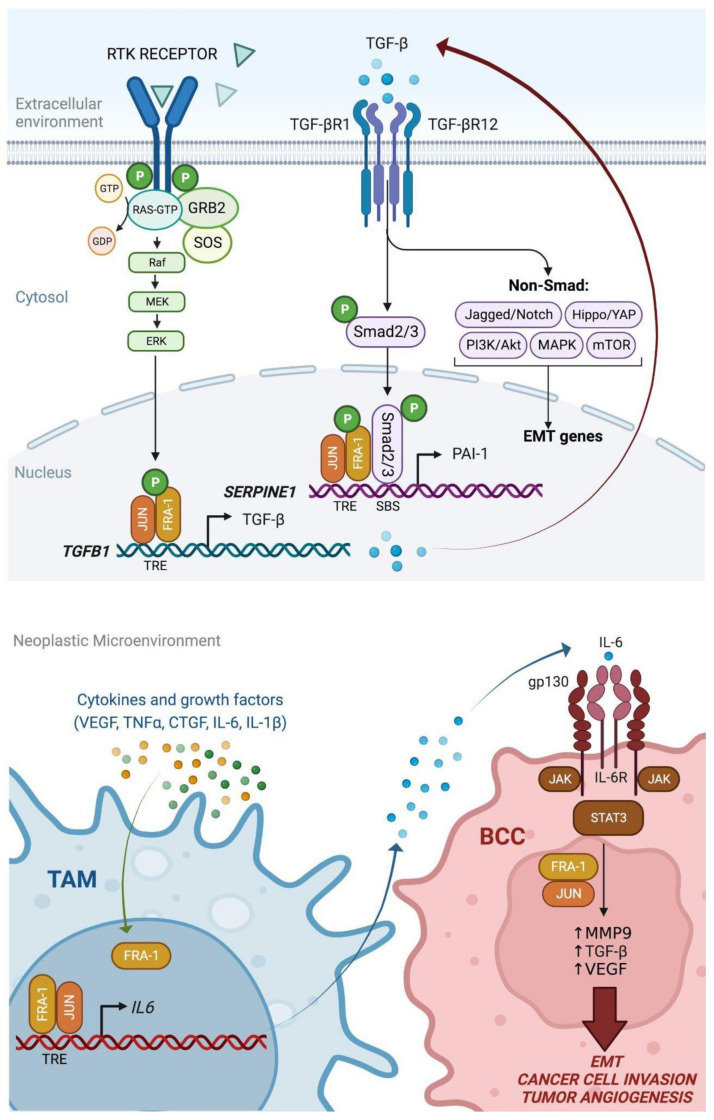
Multiple autocrine and paracrine mechanisms of FRA-1-mediated induction of TGF-beta and IL-6 in breast tumors [72]. *Upper drawing*: Dual action of FRA-1 on the TGF-beta-SMAD pathway. The FRA-1/AP-1-mediated transactivation of *TGFB1* results in the accumulation and secretion of TGF-beta, which contributes to the autocrine EMT induction in BCCs. Moreover, FRA-1 specifically interacts and cooperates with SMAD2/3 on target promoters (e.g., *SERPINE1*), thus contributing to the convergence between the RAS-MAPK-AP-1 and TGF-beta-SMAD pathways in transcriptional control of EMT. *Lower drawing*: FRA-1 contributes to the EMT paracrine mechanisms mediated by TAMs in breast cancer. In response to signals released in the neoplastic microenvironment, FRA-1 is accumulated in TAMs, in which the FRA-1/AP-1 dimers transactivate the *IL6* promoter and trigger interleukin-6 secretion. The consequent induction of the IL6-STAT3 pathway in BCCs, in cooperation with the FRA-1/AP-1 complexes, stimulates the transcription and release of MMP9, TGF-beta, and VEGF, and subsequent cancer cell invasion and mesenchymal transformation, along with increased tumor angiogenesis.

## Data Availability

Not applicable.

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
