# Peer review of "FRA-1 as a Regulator of EMT and Metastasis in Breast Cancer"

_ijms, 2023, doi:10.3390/ijms24098307_

Round 1

Reviewer 1 Report

The manuscript by Dr Verde and his colleagues discusses the biology of FRA1 protein in breast cancer in relation to epithelial mesenchymal plasticity (EMP). The authors provide comprehensive review of the literature on FRA-1 regulation, crosstalk with various EMP-related pathways, and impact on cell invasion. The paper is very well written. Although several reviews on AP-1 have been recently published, the authors discuss the topics, which have never been addressed. Of particular interest is the chapter 4 devoted to the perspectives in FRA-1 research, FRA-1 role in TAM biology and generation of hybrid E/M cells in primary tumours and in the circulation.

The paper will be of great interest for the broad community of cancer researchers.

Main issue

Throughout the text: there are no references to the Figures except for Figure 1 in the Introduction.

Minor points

1.       The references appear on page 4, but absent in the reference list (Zippo et al, 2009, 2007).

2.       Examples of gene names which must be in italic: Pages 14, 19, FOSL1; page 9 and Fig. 4 legend, CDH1; Fig. 5 legend, page 13, TGFB1; page 12, tgfb1; page 15, HMGA1; page 18, fosl1. All gene names have to be checked throughout the text.

3.       Reference 49 appears in the text after ref 28.

4.       On page 6, the NAMEC abbreviation is explained twice.

5.       It would be useful if the legends to all Figures contain references.

6.       Figure 2 has a title:  Fra-1 interactions with the EMT regulatory network and EMT-related target genes. It seems that the signalling shown under the number 6 is unrelated to this title.

7.       Figure 4. Z box/Z box in the Fig. is bipartite high affinity ZEB-binding element. Degenerated Z-box is shown on the top panel

8.       Page 12. Amuvatinib is a multikinase rather than a specific AXL inhibitor

9.       Page 13. The paper by Vial et al. 2003 Cancer Cell can be mentioned in context of integrin-uPAR signalling. Even though the experiments were performed in CRC cells, the mechanisms uncovered in this work are likely to be relevant to breast cancer cells as well.

10.   Chapter 3.3. AXL is another druggable FRA-1 target.

11.   Page 14. This statement has to be explained: “Moreover, the membrane-localized CLCA2 interacts with and downregulates the EMT-inducing beta-catenin target genes [87].”

12.   The second paragraph in section 3.4 overlaps with the discussion on pages 7 and 8 on FRA-1/EMT-TFs regulation.

13.   Page 17. One of the 9-gene classifier FRA-1 targets is EZH2 that binds ZEB1 and is involved in EMT.

14.   On page 20, the EMP (Epithelial-Mesenchymal-Plasticity) abbreviation is explained twice.

Author Response

Response to Reviewer 1 Comments

Main issue

Throughout the text: there are no references to the Figures except for Figure 1 in the Introduction.

Response : We have now added (in bold) the reference to each Figure (1 to 5) within the text.

Minor points

Point 1: The references appear on page 4, but absent in the reference list (Zippo et al, 2009, 2007).

Response 1: We have now introduced the missing articles (refs 32 and 33) in the reference list.

Point 2: Examples of gene names which must be in italic: Pages 14, 19, FOSL1; page 9 and Fig. 4 legend, CDH1; Fig. 5 legend, page 13, TGFB1; page 12, tgfb1; page 15, HMGA1; page 18, fosl1. All gene names have to be checked throughout the text.

Response 2: We have checked and corrected the gene names (in italic) throughout the text, according to the nomenclature for human vs murine genes and proteins. Moreover, we have re-checked the entire text and fixed grammar mistakes and inconsistencies.

Point 3: Reference 49 appears in the text after ref 28.

Response 3: All the citations have been re-inserted and numbered according to the order of appearance, including the mentioned reference (Basbous et al., 2007; previously numbered as 49, now ref. nr 47)

Point 4: On page 6, the NAMEC abbreviation is explained twice.

Response 4: We have eliminated the duplicated explanation for the NAMEC acronym.

Point 5: It would be useful if the legends to all Figures contain references.

Response 5: We now added the relevant citations in each Figures Legend (1 to 5).

Point 6: Figure 2 has a title: Fra-1 interactions with the EMT regulatory network and EMT-related target genes. It seems that the signaling shown under the number 6 is unrelated to this title.

Response 6: Since the signaling mechanism shown under the number 6 refers to FRA-1 and SNAIL posttranslational mechanisms, we have now changed the title of the Figure from “Fra-1 interactions with the EMT regulatory network and EMT-related target genes” to the more general “Transcriptional, posttranscriptional and posttranslational regulatory networks involving FOSL1/FRA-1, core EMT-TFs, miRNAs and FRA-1/AP-1 target genes”.

Point 7: Figure 4. Z box/Z box in the Fig. is bipartite high affinity ZEB-binding element. Degenerated Z-box is shown on the top panel.

Response 7: In response to the reviewer’s pertinent observation, we have now modified the Fig.4 and respective legend, in order to indicate the low-affinity degenerated Z-box in distal regulatory regions of ZEB1-induced genes (top part) vs the high-affinity Z box/Z box in promoter regions of ZEB1-repressed genes.

Point 8: Page 12. Amuvatinib is a multikinase rather than a specific AXL inhibitor.

Response 8: We have expanded the discussion on the AXL oncoprotein, moved from section 3.2 to 3.3 in the revised manuscript, in which we have better specified that MP470 (Amuvatinib) is a multi-targeted tyrosine kinase inhibitor, targeting AXL along with other RTKs (PDGFR, c-KIT, and MET), and introduced new references on more selective orally bioavailable AXL inhibitors.

Point 9: Page 13. The paper by Vial et al. 2003 Cancer Cell can be mentioned in context of integrin-uPAR signaling. Even though the experiments were performed in CRC cells, the mechanisms uncovered in this work are likely to be relevant to breast cancer cells as well.

Response 9: We thank the reviewer for this important remark on the seminal contribution from Chris Marshall’s group. We have now introduced a brief discussion about the functional crosstalk between FRA-1, uPAR and β1-integrin in driving cell polarization, motility, and invasiveness in human colon carcinoma cell lines and added the respective reference:

[84] E. Vial, E. Sahai, e C. J. Marshall, «ERK-MAPK signaling coordinately regulates activity of Rac1 and RhoA for tumor cell motility», Cancer Cell, vol. 4, fasc. 1, pp. 67–79, lug. 2003, doi: 10.1016/S1535-6108(03)00162-4.

Point 10: Chapter 3.3. AXL is another druggable FRA-1 target.

Response 10: We thank the reviewer for the observation. See the previous response to Point 8.

Point 11: Page 14. This statement has to be explained: “Moreover, the membrane-localized CLCA2 interacts with and downregulates the EMT-inducing beta-catenin target genes [87].”

Response 11: By expanding the discussion on the pertinent article (Ramena et al.,CLCA2 Interactor EVA1 Is Required for Mammary Epithelial Cell Differentiation [105]), we have better explained the statement as follows (Page 15): “Moreover, in the membrane CLCA2 colocalizes with E-cadherin and interacts with beta-catenin regulating homophilic cell-cell interactions while inhibiting the beta-catenin cytosolic signaling and downregulating the EMT-inducing beta-catenin target genes”. Please note that the correspondent ref number has changed from 87 to 105.

Point 12: The second paragraph in section 3.4 overlaps with the discussion on pages 7 and 8 on FRA-1/EMT-TFs regulation.

Response 12: To eliminate the redundancy between the two paragraphs, we have now minimized the reference to TRPS1 on page 9, in which we refer to the later detailed description (“as discussed below (section 3.4).”)

Point 13 Page 17. One of the 9-gene classifier FRA-1 targets is EZH2 that binds ZEB1 and is involved in EMT.

Response 13: We thank the reviewer for this important remark. We have now discussed the functional relationship between the EZH2 component of the Polycomb Repressive Complex 2, FRA-1 and lncRNAs and cited the following relative references:

[117]    C. Battistelli et al., «The Snail repressor recruits EZH2 to specific genomic sites through the enrollment of the lncRNA HOTAIR in epithelial-to-mesenchymal transition», Oncogene, vol. 36, fasc. 7, pp. 942–955, feb. 2017, doi: 10.1038/onc.2016.260.

[118]    S. Zhang et al., «LINC00152 upregulates ZEB1 expression and enhances epithelial-mesenchymal transition and oxaliplatin resistance in esophageal cancer by interacting with EZH2», Cancer Cell Int, vol. 20, fasc. 1, p. 569, dic. 2020, doi: 10.1186/s12935-020-01620-1.

Point 14: On page 20, the EMP (Epithelial-Mesenchymal-Plasticity) abbreviation is explained twice.

Response 14: We have eliminated the duplication of the EMP explanation.

Reviewer 2 Report

In the review” Fra-1 as a regulator of EMT and metastasis in breast cancer”, the authors give a comprehensive overview of the role of Fra-1 in breast cancer metastasis and EMT.

The review is well-organized and comprehensively described. The language is clear. The figures are clear, properly labeled, and well-made. Overall, the review is well-written.

Here are a few minor points the author may want to improve:

1.  In the introduction part, the author discussed the miRNA-mediated posttranscriptional regulation of EMT-TFs and other key EMT regulators. Recent studies showed that RNA binding proteins, e.g. HuR, also contribute to EMT in several types of cancer, e.g. pancreatic cancer and colon cancer. Small molecule inhibitors targeting RNA binding proteins have been proven to inhibit EMT in vitro and in vivo. I suggest the author have a paragraph to discuss the role of RNA-binding proteins in regulating EMTs. 

2. Similarly, are any known RNA-binding proteins involved in the regulation of Fra-1? Please have a brief discussion.

3. Is there any Fra-1 cre mouse model or flox mouse model, even if they are not used in cancer studies? I suggest the author have a table to list the Fra-1-related genetically modified mouse model. I could be very useful for future studies.

4. Is there any small molecule inhibitors that can directly or indirectly target Fra-1? If yes, please have a brief discussion; if no, please discuss the challenges. 

5. In addition to 4, novel AI-based approaches, e.g. alpha fold, rosettafold, have been shown to have the potential to precisely predict protein structures and design small peptide-based inhibitors. Are there any approaches to studying the structure and developing peptide-based inhibitors for Fra-1? If not, please have a brief discussion about how deep learning and AI-based approaches could help target Fra-1 as a therapeutic target.

Author Response

Response to Reviewer 2 Comments

MINOR POINTS

Point 1: In the introduction part, the author discussed the miRNA-mediated posttranscriptional regulation of EMT-TFs and other key EMT regulators. Recent studies showed that RNA binding proteins, e.g. HuR, also contribute to EMT in several types of cancer, e.g. pancreatic cancer and colon cancer. Small molecule inhibitors targeting RNA binding proteins have been proven to inhibit EMT in vitro and in vivo. I suggest the author have a paragraph to discuss the role of RNA-binding proteins in regulating EMTs.

Response 1: We thank the reviewer for indicating these important components of posttranscriptional regulation of EMT, overlooked in our original manuscript. We have now discussed these aspects in the last part of the Introduction (Section 1). Accordingly, we have included the following pertinent references:

[20]        Y. Chen, H. Qin, e L. Zheng, «Research progress on RNA−binding proteins in breast cancer», Front. Oncol., vol. 12, p. 974523, ago. 2022, doi: 10.3389/fonc.2022.974523.

[21]        I. Barbieri e T. Kouzarides, «Role of RNA modifications in cancer», Nat Rev Cancer, vol. 20, fasc. 6, pp. 303–322, giu. 2020, doi: 10.1038/s41568-020-0253-2.

[22]        J. Wu et al., «RBM38 is involved in TGF-β-induced epithelial-to-mesenchymal transition by stabilising zonula occludens-1 mRNA in breast cancer», Br J Cancer, vol. 117, fasc. 5, pp. 675–684, ago. 2017, doi: 10.1038/bjc.2017.204.

Point 2: Similarly, are any known RNA-binding proteins involved in the regulation of Fra-1? Please have a brief discussion.

Response 2: We have significantly expanded the section 2.1 (Fra-1 structure and regulation), in which we have now discussed the possible interactions of the FOSL1 transcript with IGF2BPs and other RBPs. Accordingly, we have introduced the following citations:

[39]          P. Di Fazio, «The epitranscriptome: At the crossroad of cancer prognosis», EBioMedicine, vol. 64, p. 103231, feb. 2021, doi: 10.1016/j.ebiom.2021.103231.

[40]          D. Fan et al., «Potential Target Analysis of Triptolide Based on Transcriptome-Wide m6A Methylome in Rheumatoid Arthritis», Front. Pharmacol., vol. 13, p. 843358, mar. 2022, doi: 10.3389/fphar.2022.843358.

[41]          D. Ramesh-Kumar e S. Guil, «The IGF2BP family of RNA binding proteins links epitranscriptomics to cancer», Seminars in Cancer Biology, vol. 86, pp. 18–31, nov. 2022, doi: 10.1016/j.semcancer.2022.05.009.

[42]          P. Su et al., «IMP3 expression is associated with epithelial-mesenchymal transition in breast cancer», Int J Clin Exp Pathol, vol. 7, fasc. 6, pp. 3008–3017, 2014.

[43]          A. Zirkel, M. Lederer, N. Stöhr, N. Pazaitis, e S. Hüttelmaier, «IGF2BP1 promotes mesenchymal cell properties and migration of tumor-derived cells by enhancing the expression of LEF1 and SNAI2 (SLUG)», Nucleic Acids Research, vol. 41, fasc. 13, pp. 6618–6636, lug. 2013, doi: 10.1093/nar/gkt410.

[44]          Z. Zhou et al., «HOXA11-AS1 Promotes PD-L1-Mediated Immune Escape and Metastasis of Hypopharyngeal Carcinoma by Facilitating PTBP1 and FOSL1 Association», Cancers, vol. 14, fasc. 15, p. 3694, lug. 2022, doi: 10.3390/cancers14153694.

[45]          P. Hou et al., «PTBP3-Mediated Regulation of ZEB1 mRNA Stability Promotes Epithelial–Mesenchymal Transition in Breast Cancer», Cancer Research, vol. 78, fasc. 2, pp. 387–398, gen. 2018, doi: 10.1158/0008-5472.CAN-17-0883.

Point 3: Is there any Fra-1 cre mouse model or flox mouse model, even if they are not used in cancer studies? I suggest the author have a table to list the Fra-1-related genetically modified mouse model. I could be very useful for future studies.

Response 3: We greatly appreciate the reviewer’s comment on this important aspect. To address this point, we have now expanded the section “4.2. The in vivo analysis of Fra-1 functions in mouse models of TNBC” We have better examined the opportunity to investigate the in vivo role of Fra-1 by genetic crosses between the TNBC onco-mouse models and the animals carrying the conditional knockout of Fosl1. We believe that the Table listing other genetically modified fosl1 mouse models would be beyond the scope of the present review, concerning FRA-1 in EMT in breast cancer. In addition to the strategies aimed at generating germline modifications by breeding the Fosl1-floxed lines with transgenic lines carrying mammary gland-specific Cre-recombinase, we have now discussed a highly promising approach for the rapid generation of somatic mutations by targeting the mammary gland cell populations expressing the avian viral receptor (TVA) using Replication-Competent Avian leukosis virus Splice-acceptor (RCAS) vectors (RCAS-TVA). Importantly, by this approach important results have been generated, concerning the key role of Fra-1 in driving the mesenchymal features in the aggressive glioblastoma variant (MES). Accordingly, we inserted the following references:

[130]        A. Vallejo et al., «An integrative approach unveils FOSL1 as an oncogene vulnerability in KRAS-driven lung and pancreatic cancer», Nat Commun, vol. 8, fasc. 1, p. 14294, feb. 2017, doi: 10.1038/ncomms14294.

[131]        B. Oldrini et al., «Somatic genome editing with the RCAS-TVA-CRISPR-Cas9 system for precision tumor modeling», Nat Commun, vol. 9, fasc. 1, p. 1466, apr. 2018, doi: 10.1038/s41467-018-03731-w.

[132]        Z. Du et al., «Introduction of oncogenes into mammary glands in vivo with an avian retroviral vector initiates and promotes carcinogenesis in mouse models», Proc. Natl. Acad. Sci. U.S.A., vol. 103, fasc. 46, pp. 17396–17401, nov. 2006, doi: 10.1073/pnas.0608607103.

[133]        C. Marques et al., «NF1 regulates mesenchymal glioblastoma plasticity and aggressiveness through the AP-1 transcription factor FOSL1», eLife, vol. 10, p. e64846, ago. 2021, doi: 10.7554/eLife.64846.

Point 4: Is there any small molecule inhibitors that can directly or indirectly target Fra-1? If yes, please have a brief discussion; if no, please discuss the challenges.

Response 4: We thank the reviewer for these important remarks. Accordingly, we have introduced a new (final) section 4.6 (FOSL1/FRA-1 as a target of therapeutic intervention in TNBC/basal-like breast cancer), in which we have discussed the main strategies for the inhibition of FRA-1 expression and/or activity. Therefore, we took the opportunity to cite and summarize our recent review article, entitled “The Fra-1/AP-1 Oncoprotein: From the “Undruggable” Transcription Factor to Therapeutic Targeting», specifically focused on translational approaches for (direct or indirect) therapeutic targeting of Fra-1.

[55]          L. Casalino, F. Talotta, A. Cimmino, e P. Verde, «The Fra-1/AP-1 Oncoprotein: From the “Undruggable” Transcription Factor to Therapeutic Targeting», Cancers, vol. 14, fasc. 6, p. 1480, mar. 2022, doi:

Moreover, we cited and added the references to the following articles, which also include one recent report concerning a promising FRA-1-specific peptide inhibitor:

[158]        J. G. Albeck, G. B. Mills, e J. S. Brugge, «Frequency-Modulated Pulses of ERK Activity Transmit Quantitative Proliferation Signals», Molecular Cell, vol. 49, fasc. 2, pp. 249–261, gen. 2013, doi: 10.1016/j.molcel.2012.11.002.

[159]        A. V. Vaseva et al., «KRAS Suppression-Induced Degradation of MYC Is Antagonized by a MEK5-ERK5 Compensatory Mechanism», Cancer Cell, vol. 34, fasc. 5, pp. 807-822.e7, nov. 2018, doi: 10.1016/j.ccell.2018.10.001.

[160]        P. Guo, J. Yang, J. Huang, D. T. Auguste, e M. A. Moses, «Therapeutic genome editing of triple-negative breast tumors using a noncationic and deformable nanolipogel», Proc. Natl. Acad. Sci. U.S.A., vol. 116, fasc. 37, pp. 18295–18303, set. 2019, doi: 10.1073/pnas.1904697116.

[161]        M. Yu, L. Ghamsari, J. A. Rotolo, B. J. Kappel, e J. M. Mason, «Combined computational and intracellular peptide library screening: towards a potent and selective Fra1 inhibitor», RSC Chem. Biol., vol. 2, fasc. 2, pp. 656–668, 2021, doi: 10.1039/D1CB00012H.

Point 5: In addition to 4, novel AI-based approaches, e.g. alpha fold, rosettafold, have been shown to have the potential to precisely predict protein structures and design small peptide-based inhibitors. Are there any approaches to studying the structure and developing peptide-based inhibitors for Fra-1? If not, please have a brief discussion about how deep learning and AI-based approaches could help target Fra-1 as a therapeutic target.

Response 5: We thank the reviewer for suggesting how these new bioinformatic methods pave the way for innovative studies on FRA-1 therapeutic targeting. In the new section (4.6: FOSL1/FRA-1 as a target of therapeutic intervention in TNBC/basal-like breast cancer), after examining the results and ongoing research on chemical compounds and polypeptides inhibitors targeting the Fra-1/AP-1 heterodimers, we refer to the forthcoming application of the powerful AI-based computational methods. In particular, we envisage how the recently developed structure prediction tools can be exploited for the in silico design of small selective inhibitors of the FRA-1/AP-1 activity. Accordingly, the following citations have been included:

[162]        J. Jumper et al., «Highly accurate protein structure prediction with AlphaFold», Nature, vol. 596, fasc. 7873, pp. 583–589, ago. 2021, doi: 10.1038/s41586-021-03819-2.

[163]        M. Varadi et al., «AlphaFold Protein Structure Database: massively expanding the structural coverage of protein-sequence space with high-accuracy models», Nucleic Acids Research, vol. 50, fasc. D1, pp. D439–D444, gen. 2022, doi: 10.1093/nar/gkab1061.

[164]        M. Baek et al., «Accurate prediction of protein structures and interactions using a three-track neural network», Science, vol. 373, fasc. 6557, pp. 871–876, ago. 2021, doi: 10.1126/science.abj8754.

[165]        I. R. Humphreys et al., «Computed structures of core eukaryotic protein complexes», Science, vol. 374, fasc. 6573, p. eabm4805, dic. 2021, doi: 10.1126/science.abm4805.
